# Differential but complementary roles of HIF-1α and HIF-2α in the regulation of bone homeostasis
Sun Young Lee[1,4], Su-Jin Kim[1,2,4], Ka Hyon Park[1,2], Gyuseok Lee [1], Youngsoo Oh[1], Je-Hwang Ryu [1,2] ✉ & Yun Hyun Huh [3] ✉

Bone is a highly dynamic tissue undergoing continuous formation and resorption. Here, we investigated differential but complementary roles of hypoxia-inducible factor (HIF)-1α and HIF-2α in regulating bone remodeling. Using RNA-seq analysis, we identified that specific genes involved in regulating osteoblast differentiation were similarly but slightly differently governed by HIF-1α and HIF-2α. We found that increased HIF-1α expression inhibited osteoblast differentiation via inhibiting RUNX2 function by upregulation of *Twist2*, confirmed using *Hif1a* conditional knockout (KO) mouse. Ectopic expression of HIF-1α via adenovirus transduction resulted in the increased expression and activity of RANKL, while knockdown of *Hif1a* expression via siRNA or osteoblast-specific depletion of *Hif1a* in conditional KO mice had no discernible effect on osteoblast-mediated osteoclast activation. The unexpected outcome was elucidated by the upregulation of HIF-2α upon *Hif1a* overexpression, providing evidence that *Hif2a* is a transcriptional target of HIF-1α in regulating RANKL expression, verified through an experiment of HIF-2α knockdown after HIF-1α overexpression. The above results were validated in an ovariectomized- and aging-induced osteoporosis model using *Hif1a* conditional KO mice. Our findings conclude that HIF-1α plays an important role in regulating bone homeostasis by controlling osteoblast differentiation, and in influencing osteoclast formation through the regulation of RANKL secretion via HIF-2α modulation.

Bone is a dynamic tissue characterized by old matrix resorption by osteoclasts and new matrix formation by osteoblasts. The balance between bone formation and bone resorption is essential for bone homeostasis. Osteoblasts, originating from mesenchymal stem cells, are controlled by osteogenic transcription factors, runt-related transcription factor (RUNX2)[1], osterix[2], and TCF/LEF[3]. Osteoclasts are derived from hematopoietic progenitors of the monocyte-macrophage lineage[4]. The initial proliferation of pre-osteoclasts is controlled by macrophage colony-stimulating factor (M-CSF), and osteoclast differentiation and maturation are governed by receptor activator of nuclear factors (NF)-κB ligand (RANKL)[5]. RANKL is expressed in the osteoblast-lineage cells and stimulates RANK, the receptor for RANKL, on osteoclast precursors. Interaction of RANKL on osteoblasts and RANK on osteoclasts results in the activation of signal transduction for the differentiation and maturation of osteoclasts, such as NF-κB, c-Fos, and nuclear factor of

activated T cells 1 (NFATc1)[6,7]. Osteoprotegerin (OPG) produced by osteoblasts is a soluble decoy receptor for RANKL and regulates RANKL-mediated osteoclastogenesis[8].

Hypoxia-inducible factor (HIF) is a heterodimeric transcription factor consisting of an inducible α subunit and constitutive β subunit. HIF-α is oxygen labile, and HIF-α stability is crucial for the activity of HIF as a transcriptional regulator[9]. Under normoxia, the proline residues of the HIF-α subunit are hydroxylated by prolyl-4-hydrolyase (PHD) following interaction with von Hippel-Landau (VHL), specific E3 ligase, and then polyubiquitination and proteasomal degradation[10]. However, in hypoxia, proline hydroxylation is prohibited by the deprivation of oxygen, allowing the accumulation of HIF-α and the formation of an active heterodimeric complex with HIF-1β[11]. Remarkably, HIF-α can be stabilized even in normoxia by inflammatory factors, such as IL-1β, INF-γ, IGF-1 and TNF-α, in a pathophysiologic microenvironment[12,13], and

[1]Department of Pharmacology and Dental Therapeutics, School of Dentistry, Chonnam National University, Gwangju, 61186, Republic of Korea. [2]Research Center for Biomineralization Disorders, School of Dentistry, Chonnam National University, Gwangju, 61186, Republic of Korea. [3]School of Life Sciences, Gwangju Institute of Science and Technology, Gwangju, 61005, Republic of Korea. [4]These authors contributed equally: Sun Young Lee, Su-Jin Kim. ✉e-mail: jesryu@jnu.ac.kr; yhuh@gist.ac.kr

normoxic stabilization of HIF-1α induces osteoclastogenesis and pathological bone resorption[14].

Of the three identified isotypes of HIF-α, studies have focused on HIF-1α and HIF-2α. They retain highly conserved sequences, have similar domain structures, and bind to the same hypoxia-responsive element (HRE), 5′-(A/G)CGTG-3′ sequence, in promoters of specific target genes[15,16]. Studies show that the effects of HIF-1α and HIF-2α in regulating skeletal development, bone remodeling, and pathological bone diseases are different. Some papers suggest the anabolic role of HIF-α. Accumulation of HIF-1α and HIF-2α in osteoblasts via specific deletion of VHL promotes endochondral ossification and long bone formation by promoting vascularization[17,18]. On the contrary, other studies support the catabolic effects of HIF-1α. Deletion of HIF-1α in osteoblasts of mature bone resulted in the increased accumulation of bone[19]. Overexpression of HIF-1α negatively regulates mechanical load-induced bone formation[20] and inhibits the proliferation and growth of osteoblasts by synergistic inhibition of the Wnt pathway and osterix[21]. Bone is a hypoxic tissue with different regions, such as cortical bone, bone marrow, cancellous bone, and endosteum, characterized by different hypoxia levels depending on the distribution of large or small blood vessels[22]. The effects of hypoxia on skeletal development and bone remodeling are complex and differ with the early or late stage of differentiation and diverse oxygen levels[22]. Therefore, the effects of HIF-1α and HIF-2α involved in skeletal development, bone remodeling, and pathologic bone disease would differ.

Our previous study observed the catabolic functions of HIF-2α in maintaining bone homeostasis and normoxic stabilization of HIF-1α and HIF-2α when cultured in differentiation-inducing conditions[23]. Here, we demonstrate the role of normoxic stabilization of HIF-1α in regulating bone remodeling, and by comparing with HIF-2α function, we suggest the differential but complementary regulation of HIF-1α and HIF-2α.

## Results
### Transcriptional profiles regulated by HIF-1α and HIF-2α during osteoblast differentiation
Our previous study observed the accumulation of HIF-1α and HIF-2α during in vitro osteoblast differentiation under normoxia when pre-osteoblasts were cultured in differentiation media (DM)[23]. To determine the distinct roles of HIF-1α in regulating bone homeostasis, at first, we confirmed the expression and normoxic stabilization of HIF-1α and HIF-2α during osteoblast differentiation. Expression of *Hif1a* was significantly increased in the early stage of osteoblast differentiation before the increase of *Ocn*, *Alp*, and *Runx2* expression, the markers of osteoblasts, which is a similar pattern of *Hif2a* expression (Fig. 1a, Supplementary Fig. 1a). Increased accumulation and nuclear localization of HIF-1α and HIF-2α were observed on day 6 with the media containing differentiation-inducing agents, such as L-AA and β-Gp (Fig. 1b). Expression of HIF-1α in the nucleus persisted from day 6 to day 12 of differentiation, while HIF-2α increased until day 15 (Supplementary Fig. 1b). RNA-seq analysis showed similar transcriptional profiles in primary calvarial pre-osteoblasts by overexpression of HIF-1α or HIF-2α (Fig. 1c). As a result of RNA-seq analysis, 838 (62.7%) differentially expressed genes (DEGs) (out of 40,213 genes) were shared, and 146 (10.9%) and 353 (26.4%) DEGs were unique to HIF-1α and HIF-2α, respectively (Fig. 1d). To comprehend the complex cellular processes and functions associated with HIF transcriptional regulation, we clustered the DEGs based on fold enrichment and selected the most informative Gene Ontology (GO) terms for each set of HIF DEGs. Fig. 1e shows the top enriched cellular processes, including extracellular signal-regulated kinase (ERK)-1/-2 cascade, immune response, and positive regulation of cell migration strongly shared between HIF-1α and HIF-2α. Of them, 3905 genes involved in cell differentiation were analyzed, and we found that most were similarly regulated by HIF-1α and HIF-2α (Fig. 1f). Representatively, well-known effectors such as *Ptgs2*, *Fas*, *Il6*, *Rankl*, *Twist2*, *Vegfa*, and *Vegfc* were upregulated and *Bglap1*, *Bglap2*, *Bmp2*, *Fgfr2*, *Fgfr3*, *Sp7*, and *Pparg* were down-regulated (Fig. 1f). Thus, we determined that about 62.7% of filtered genes were commonly regulated;

however, some genes were distinctly regulated by HIF-1α and HIF-2α during osteoblast differentiation under normoxia.

### HIF-1α inhibits osteoblast differentiation via the TWIST2-RUNX2-OCN axis
Ectopic expression of HIF-1α via the adenovirus transduction system showed inhibited differentiation of primary calvarial pre-osteoblasts. The expression of osteoblast marker genes, *Ocn* and *Runx2*, was reduced by HIF-1α overexpression (Fig. 2a), and mineralization and calcified nodule formation were inhibited as determined by ALP and ARS staining (Fig. 2b). The inhibitory effect of HIF-1α on osteoblast differentiation was confirmed by BMP-2-induced regeneration of calvarial bone defect models (Fig. 2c). Implantation with Ad-*Hif1a* in the calvarial defect regions considerably delayed BMP-2-induced bone regeneration (Fig. 2c). Knockdown of *Hif1a* via RNAi increased osteoblast marker gene expression (Fig. 2d), mineralization, and calcified nodule formation (Fig. 2e). Osteocalcin (OCN) encoded by *Ocn* is an essential factor for bone matrix mineralization, and RUNX2 is a well-known transcriptional regulator of *Ocn*. RUNX2 activity as a transcription factor was determined using two types of RUNX2-responsive luciferase reporters (6XOSE-*Luc* and OG2-*Luc*), and the results show that RUNX2 activity was significantly decreased by HIF-1α overexpression (Fig. 2f). Recently, we reported that HIF-2α inhibits osteoblast differentiation via the TWIST2-RUNX2-OCN axis[23]. In our current RNA-seq data (Fig. 1f), *Twist2* was commonly upregulated by HIF-1α and HIF-2α overexpression. Similar to the effects of HIF-2α[23], *Twist2*, but not *Twist1*, was upregulated by HIF-1α overexpression (Fig. 2g). Moreover, we determined that the regulatory cis-elements of *Twist2* promoter contained putative HIF-1α binding sequences, 5′-(A/G)CGTG-3′, and ChIP results showed that *Twist2* is a direct target of HIF-1α (Fig. 2h, Supplementary Fig. 3). Additionally, siRNA-mediated silencing of *Hif1a* in primary cultured calvarial osteoblasts confirmed that *Hif1a* regulates *Twist2* expression (Fig. 2i). *Twist2* knockdown with specific siRNA partially restored the decrease of *Ocn* and *Runx2* by overexpression of *Hif1a* (Fig. 2j). These data show that HIF-1α and HIF-2α have similar effects on osteoblast differentiation by regulating the TWIST2-RUNX2-OCN axis.

### Osteoblast-specific conditional KO of *Hif1a* increases bone mass
To determine the direct role of HIF-1α in regulating bone homeostasis, we generated osteoblast-specific *Hif1a*-deficient mice by crossing *Hif1a*<sup>fl/fl</sup> mice with *Col1a1-Cre* transgenic mice. Immunohistochemical staining data verified specific knockout of *Hif1a* in osteoblasts, but not in osteoclasts, of *Hif1a*<sup>fl/fl</sup>;*Col1a1-Cre* (*Hif1a* conditional KO) mice (Fig. 3a). The three-dimensional (3D) microarchitecture of femoral trabecular bones in 4-month-old *Hif1a* conditional KO mice and their control *Hif1a*<sup>fl/fl</sup> mice was analyzed using μCT. The μCT images exhibited an increase in cancellous trabeculae (Fig. 3b), supported by quantitative bone parameters, such as BMD, BV/TV, Tb.Th, Tb.Sp, and Tb.N (Fig. 3c). The BMD, BV/TV, Tb.Th, and Tb.N showed higher values, and Tb.Sp exhibited lower values in *Hif1a*<sup>fl/fl</sup>;*Col1a1-Cre* mice than in *Hif1a*<sup>fl/fl</sup> control mice (Fig. 3c). To complement the 3D-μCT data, we conducted histomorphometric analysis of morphometric parameters of osteoblasts in the metaphyseal regions using H&E staining. It also revealed that the bone parameters, BV/TV, N.Ob/B.Pm, and Ob.S/BS, showed higher values in *Hif1a*<sup>fl/fl</sup>;*Col1a1-Cre* mice (Fig. 3d). Taken together, HIF-1α is a negative regulator in osteoblast differentiation.

### HIF-1α indirectly promotes RANKL-mediated osteoclastogenesis via HIF-2α
The regulation of osteoblast-mediated osteoclastogenesis via the regulatory role of RANKL and OPG, a decoy receptor for RANKL, is essential to maintain bone homeostasis[8]. To determine whether the inhibitory function of HIF-1α on osteoblast differentiation affects the crosstalk between osteoblasts and osteoclasts. Expression of *Rankl* and *Opg* by *Hif1a*-overexpressing osteoblasts was examined for RANKL-mediated osteoclast maturation. HIF-1α overexpression significantly increased *Rankl* expression (but not *Opg*) in

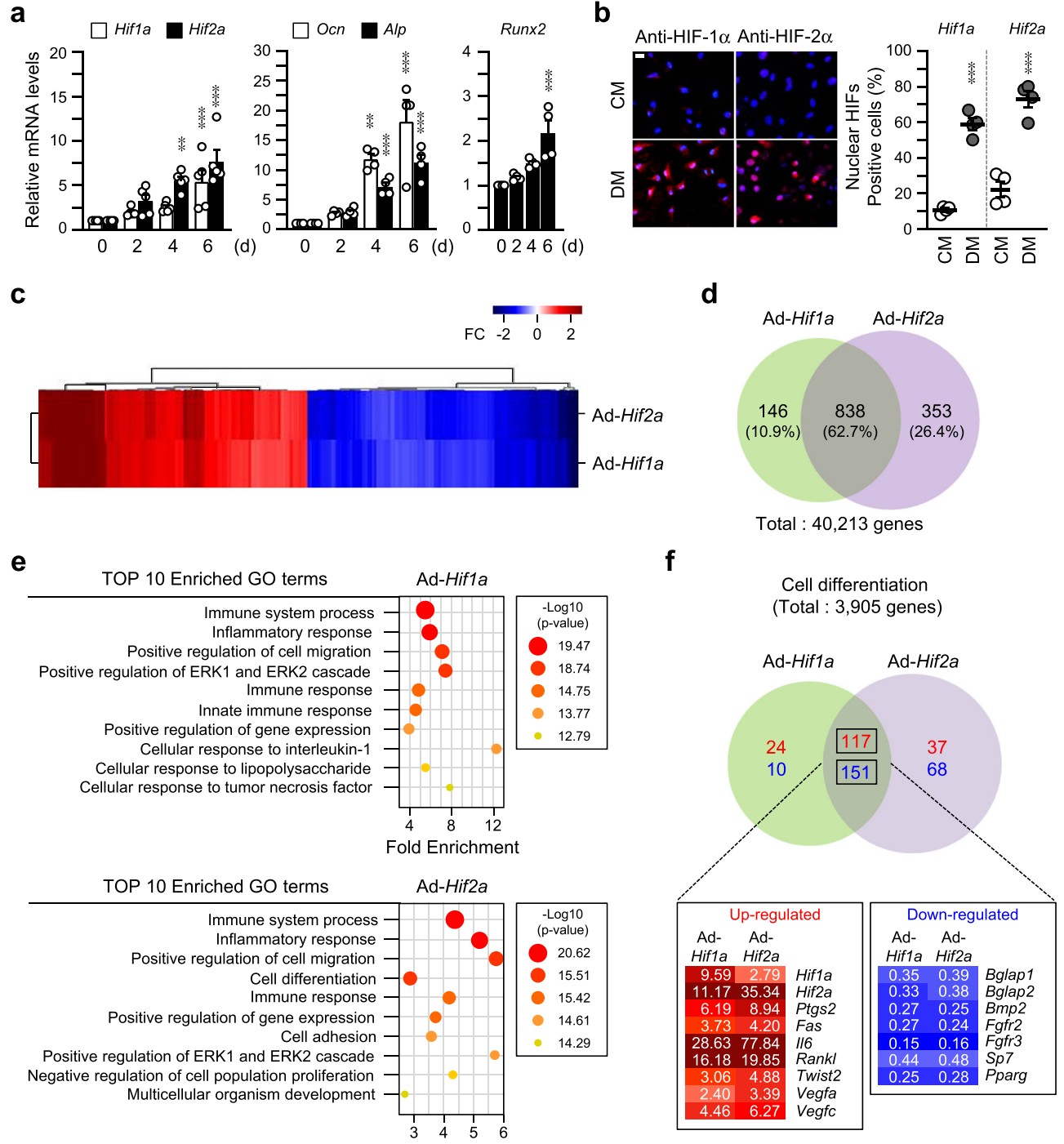

**Fig. 1 | Transcriptional profiles regulated by HIF-1α and HIF-2α during osteoblast differentiation. a** Primary calvarial pre-osteoblasts from WT mice were cultured in the medium containing 50 μg/ml L-AA and 5 mM β-Gp for up to 6 days for osteogenic differentiation. Relative transcript levels of *Hif1a*, *Hif2a*, *Ocn*, *Alp*, and *Runx2* on the indicated culture days were determined using qRT-PCR ($n \geq 4$). **b** At day 6 differentiation, nuclear translocation of HIF-1α and HIF-2α in osteogenic differentiation medium (DM) was observed using immunocytochemistry with anti-HIF-1α antibody, anti-HIF-2α antibody and Alex-594 (red) ($n = 4$). Nuclei were marked with DAPI (blue) staining. CM, control media. Scale bar, 25 μm. **a, b** Values are presented as the mean ± SEM. \*\*$P < 0.01$, and \*\*\*$P < 0.001$. **c–f** Primary cultured calvarial pre-osteoblasts were transduced with Ad-*Hif1a* or Ad-*Hif2a* on day 3, and then cultured in osteogenic DM until day 6. Total RNAs were extracted and then used for RNA-seq analysis ($n = 3$). Hierarchical clustering heatmap for DEGs was visualized (**c**). Venn diagram for DEGs of RNA-seq analysis (**d**). GO enrichment analysis (**e**). Top 10-ranked enriched GO terms were listed in the bubble chart. Enriched GO terms by *Hif1a* (top) or *Hif2a* overexpression (bottom). Venn diagram of selected DEGs associated with cell differentiation was visualized and the representative genes were listed. Absolute value of fold change (FC) > 2, normalized data (log2) > 4 (**d**, **e**) or normalized data (log2) > 2 (**f**), and p-value (paired t-test) < 0.05 were used as the cut-offs (**f**).

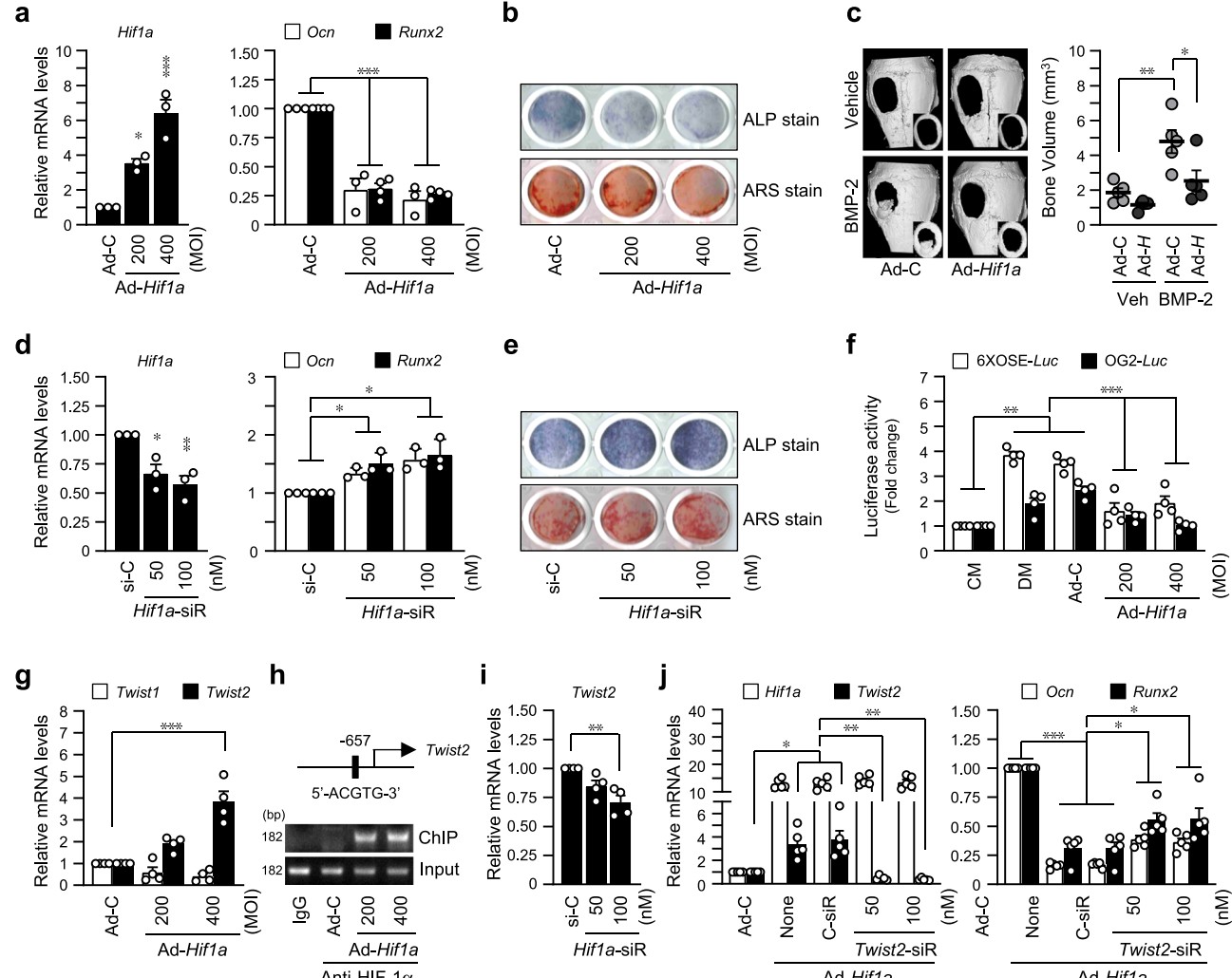

**Fig. 2 | HIF-1α inhibits osteoblast differentiation via the TWIST2-RUNX2-OCN axis. a** Osteogenic differentiation of calvarial pre-osteoblasts was induced by DM, and 200 or 400 MOI of Ad-*Hif1a* was infected at differentiation-inducing culture day 3. Subsequently, cells were harvested on the 6th day of differentiation. Relative mRNA levels of *Hif1a*, *Ocn*, and *Runx2* were quantitated by qRT-PCR (*n* ≥ 3). **b** Osteogenic phenotypes were determined by ALP and ARS staining. The representative captured images of 24-well plates were displayed (*n* = 3). **c** 5-mm diameter critical-sized defects were created in 6-week-old male mice and covered with collagen sponges without (Veh) or with 300 ng BMP-2. For each group, collagen sponges containing Ad-C or Ad-*Hif1a* were applied. After 2 weeks, the size and bone volume of calvarial defects were measured. The representative μCT images were shown (*n* = 5). **d** Cells were transfected with siRNA against *Hif1a* or control-

siRNA (si-C) on differentiation day 3 and cultured for 3 days. Transcript levels of *Hif1a*, *Ocn*, and *Runx2* were determined by qRT-PCR (*n* = 3). **e** The representative images of ALP and ARS staining (*n* = 3). **f** The pre-osteoblasts were co-transfected with RUNX2-responsive luciferase reporter constructs (6XOSE-*Luc* or OG2-*Luc*) and infected together with indicated MOI of Ad-*Hif1a*. **g** The relative RNA levels of *Twist1* and *Twist2* isotypes in *Hif1a*-overexpressing cells (*n* = 4). **h** ChIP analysis showing the HIF-1α binding to the *Twist2* promoter region. **i** mRNA level of *Twist2* in pre-osteoblasts transfected with *Hif1a* siRNA (*n* = 4). **j** Cells were transfected with siRNA against *Twist2* or si-C following infection of Ad-*Hif1a*. Transcript levels of *Hif1a*, *Twist2*, *Ocn*, and *Runx2* were analyzed (*n* = 5). Values are presented as the mean ± SEM. **P* < 0.05, ***P* < 0.01, and ****P* < 0.001.

osteoblasts (Fig. 4a). *Rankl* to *Opg* ratio at the transcript level (Fig. 4b), immunofluorescence staining of RANKL proteins (Fig. 4c), and secreted level of RANKL (Fig. 4d) confirmed upregulation of RANKL by HIF-1α in osteoblasts. However, unexpectedly, specific siRNA-mediated silencing of *Hif1a* did not affect *Rankl* expression (Fig. 4e). Furthermore, TRAP staining of trabecular bones of *Hif1a*[fl/fl];*Col1a1-Cre* mice showed no differences compared to *Hif1a*[fl/fl] control mice (Fig. 4f). These results led to the hypothesis that HIF-1α-mediated *Rankl* upregulation may be indirect. Based on the results of Fig. 1f, we compared the expression of *Hif1a* and *Hif2a* by adenovirus mediated overexpression of each isoform of HIFs. Consistent with the RNA-seq analysis, *Hif2a* expression was significantly increased by HIF-1α overexpression, whereas *Hif1a* was slightly increased by HIF-2α overexpression, while ectopic expression of each isoform of HIF-α showed an increase in *Rankl* expression (Fig. 5a, Supplementary Fig. 3). Promoter analysis of *Hif2a* revealed the existence of putative HIF-α binding sequence,

and direct binding of HIF-1α to this regulatory element of *Hif2a* promoter was determined by ChIP assay, suggesting that *Hif2a* is a direct target gene of HIF-1α (Fig. 5b, Supplementary Fig. 3). Interestingly, silencing of HIF-2α blocked Ad-*Hif1a*-mediated *Rankl* expression (Fig. 5c), whereas knockdown of HIF-1α did not affect Ad-*Hif2a*-mediated *Rankl* expression (Fig. 5d), indicating that HIF-1α-induced *Rankl* expression was mediated by HIF-2α. We set up the co-culture system to examine the effects of indirect regulation of *Rankl* expression by the HIF-1α-to-HIF-2α axis on osteoblast-mediated osteoclastogenesis. Calvarial pre-osteoblasts and BMMs, precursor cells of osteoclasts, were co-cultured in the same culture dish, and osteoblast differentiation was induced by L-AA, β-Gp, and VitD₃ and then osteoclast differentiation was measured by TRAP staining (Fig. 5e). Osteoclastogenesis of co-cultured BMMs with pre-osteoblasts transduced with Ad-*Hif2a* was more enhanced than co-culture with HIF-1α overexpressing cells. This was further evidenced by an increase in the number of TRAP-positive

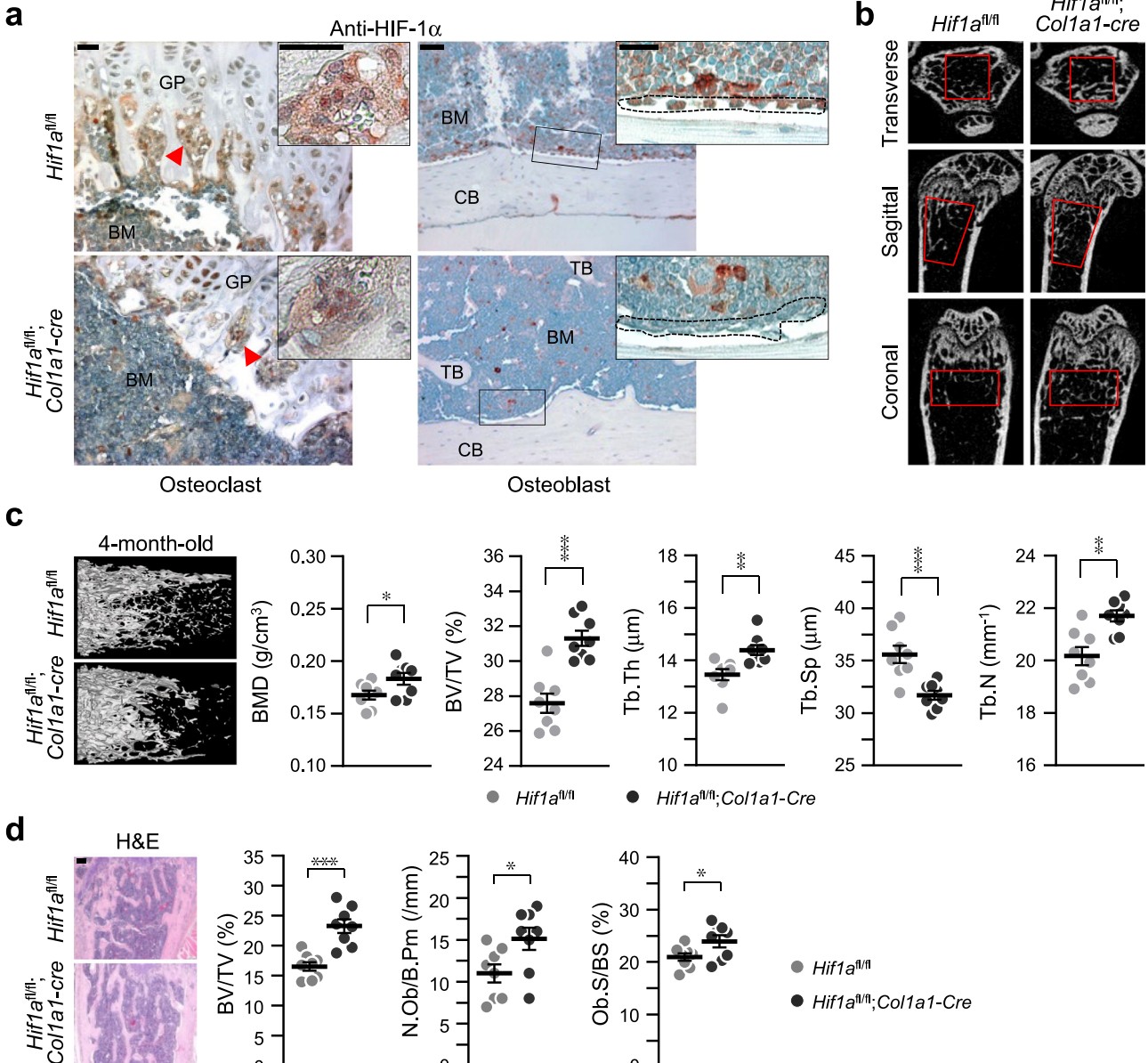

**Fig. 3 | Osteoblast-specific depletion of *Hif1a* increases bone mass. a** Osteoblast-specific, but not osteoclasts, depletion of *Hif1a* in *Hif1a*^fl/fl;*Col1a1-Cre* mice and their wild-type littermates, *Hif1a*^fl/fl was verified by immunohistochemistry with anti-HIF-1α antibody (*n* = 3). The arrows indicate osteoclasts, and the dotted line indicates osteoblasts. GP, growth plate; BM, bone marrow; TB, trabecula bone; CB, cortical bone. Scale bar, 25 μm. **b** μCT images of femurs captured by the transverse (top), sagittal (middle), and coronal (bottom) planes. Red boxes indicate regions of interest. **c** Representative images of μCT reconstructions of trabecular bones and quantitative analyses of BMD, BV/TV, Tb.Th, Tb.Sp, and Tb.N of femora (*n* = 8). **d** H&E staining results and bone histomorphometric analysis, including BV/TV, N.Ob/B.Pm, and Ob.S/BS (*n* = 8). Scale bar, 100 μm. Values are presented as the mean ± SEM. Unpaired *t*-test was performed. *$P < 0.05$, **$P < 0.01$, and ***$P < 0.001$.

multinucleated cells (Fig. 5e). In terms of regulating osteoblast differentiation, specific silencing of *Hif2a* restored Ad-*Hif1a*-mediated reduction of *Ocn* and *Runx2* expression (Fig. 5f) and knockdown of HIF-1α also recovered Ad-*Hif2a*-mediated inhibition of osteoblast marker gene expression (Fig. 5g). These data suggest a differential but complementary role of HIF-1α and HIF-2α in regulating RANKL-mediated osteoclastogenesis, although the effects of both isoforms of HIF-α were similar on osteoblast differentiation.

**Osteoblast-specific depletion of *Hif1a* delayed osteoporotic phenotypes**

To examine the effects of HIF-1α on osteoporotic bone loss, OVX (Fig. 6a–c) and age-related (Fig. 6d–f) experimental animal models were applied. μCT

images exhibited delay of OVX-induced bone loss in *Hif1a*^fl/fl;*Col1a1-Cre* mice compared to *Hif1a*^fl/fl control mice, supported by the parameters of quantitative analyses, such as BMD, BV/TV, Tb.Th, Tb.Sp, and Tb.N (Fig. 6a). Bone histomorphometric analyses with H&E (Fig. 6b) and TRAP staining results (Fig. 6c) revealed that osteoblast-specific depletion of *Hif1a* prevented OVX-induced reduction of osteoblasts (Fig. 6b) but not OVX-induced bone resorption (Fig. 6c). In particular, osteoblast-specific depletion of *Hif1a* reduced TWIST2 expression in osteoblasts of the OVX model, while the effect of *Hif1a* deficiency on HIF-2α and RANKL expression was not notably significant (Supplementary Fig. 2). Consistent with the data from OVX experimental mice, aging (12-month-old)-induced osteoporosis was also delayed in *Hif1a*^fl/fl;*Col1a1-Cre* mice (Fig. 6d–f). Taken all together, we concluded that HIF-1α is a potent regulator of bone homeostasis, similar

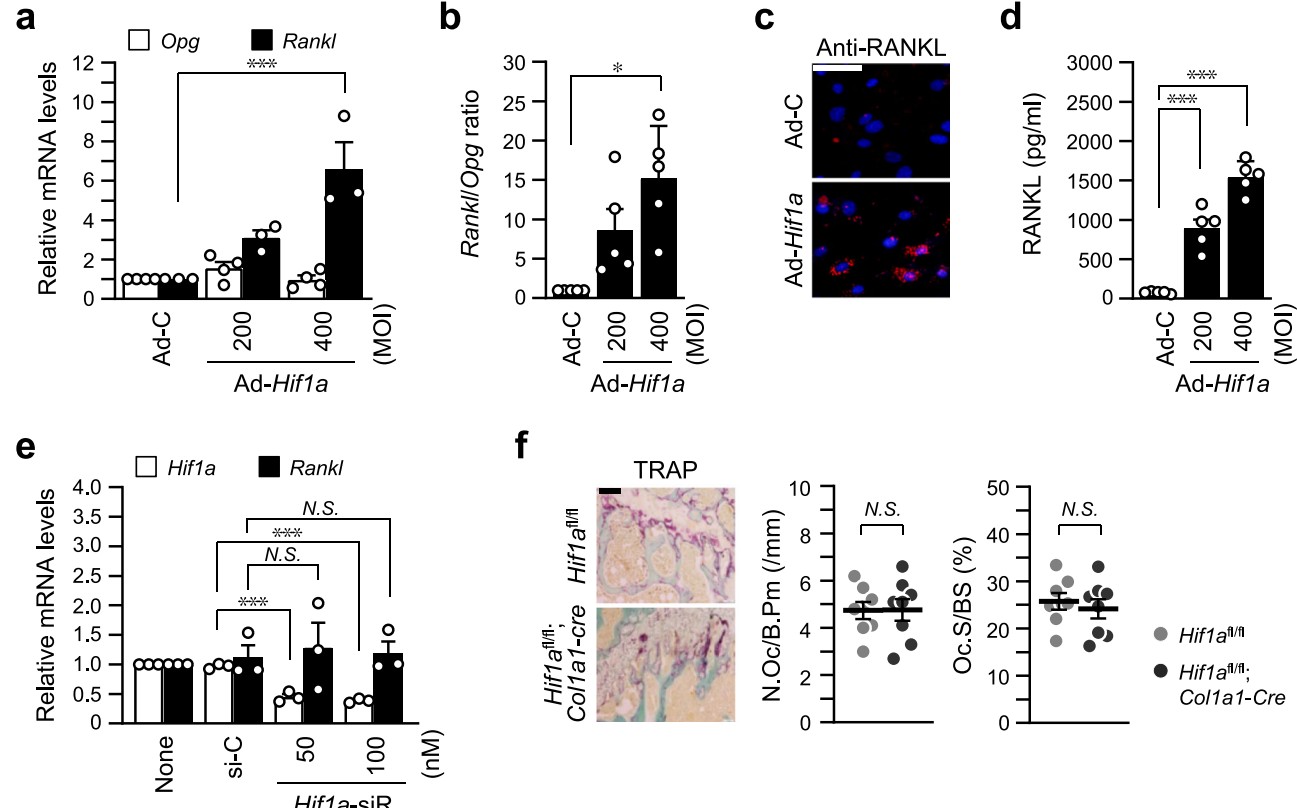

**Fig. 4 | HIF-1α increases RANKL expression, but *Hif1a*-KO does not affect osteoclast maturation. a**, **b** Primary pre-osteoblasts were infected with Ad-C or Ad-*Hif1a* on differentiation day 3 and cultured for 3 days. Relative mRNA levels of *Opg* and *Rankl* (**a**) and the *Rankl*/*Opg* ratio (**b**) were quantitated ($n \geq 3$). **c** RANKL expression at the protein level was examined by immunofluorescence staining with anti-RANKL antibody and Alexa-594 (red), and nuclei were stained with DAPI (blue) ($n = 3$). Scale bar, 100 μm. **d** Secreted RANKL was measured by ELISA using a culture medium ($n = 5$). **e** qRT-PCR of *Hif1a* and *Rankl* following siRNA-mediated silencing of *Hif1a* during osteogenic differentiation ($n = 3$). **f** Bone histomorphometric analysis of osteoclast parameters such as N.Oc/B.Pm and Oc.S/BS after TRAP staining ($n = 8$). Scale bar, 100 μm. Values are presented as the mean ± SEM. N.S. not significant, *$P < 0.05$, and ***$P < 0.001$.

to HIF-2α, but HIF-1α acts as an upstream of HIF-2α in regulating RANKL-mediated osteoclastogenesis.

## Discussion

Bone is a highly dynamic tissue that continuously undergoes remodeling to reflect an equilibrium of functional and metabolic demands even after bone development. Bone diseases, such as osteoporosis, osteopetrosis, and fragile bone fracture, are mainly caused by disturbances in bone homeostasis maintained by a balance between bone-forming osteoblasts and bone-resorbing osteoclasts. Our previous study demonstrated the critical role of HIF-2α as a catabolic regulator in bone remodeling and HIF-2α as a novel intrinsic mediator of age-related bone loss[24], which led to the investigation of HIF-1α functions in regulating bone homeostasis. The activities of the HIF-α/β dimer on the transcriptional regulation of target genes depend on the stability of the HIF-1α subtype. So far, three isotypes of HIF-α have been identified; of these, HIF-1α and HIF-2α exhibit nearly the same amino acid sequence homology, while HIF-3α has relatively short sequences. Since both HIF-1α and HIF-2α have very similar structures and can recognize and bind the same HRE, 5′-(A/G)CGTG-3′, sequences in a promoter of specific target genes, many papers have reported similar functions of these two isotypes of HIF-α. However, several studies have supported distinct roles of HIF-1α and HIF-2α in different cellular processes. Such reports show that HIF-1α is associated with acute hypoxic response, whereas HIF-2α is related to chronic hypoxic response[25]. Moreover, HIF-1α contributes to the maintenance of chondrocyte phenotypes and metabolic adaptation to the hypoxic environment, whereas HIF-2α accelerates osteoarthritic processes[26]. This study determined similar and independent regulation of HIF-1α and HIF-2α during osteoblast differentiation but differential

regulation of HIF-1α depending on HIF-2α in osteoclast activation via crosstalk with osteoblasts.

It is well-known that HIF-α stability is very sensitive to oxygen. HIF-α is hydroxylated by PHD[27] and acetylated by ARD-1[28], resulting in the proteasomal degradation of HIF-α under normoxia. However, some investigations have reported that normoxic HIF-1α stabilization is caused by local inflammatory factors, such as TNF-α, INF-γ, IL-1β, and IGF-1, in human coronary endothelial cells[13], and nitric oxide also promotes normoxic HIF-1α stabilization by inhibition of PHD[29]. Consistent with this finding, inflammatory cytokines, such as IL-1β, IL-17, IL-21, and TNF-α, induced increased expression of HIF-2α in mouse articular chondrocytes under normoxia[30]. The effect of inflammation on bone is mediated by inflammatory cytokines, such as IL-1β, IL-6, IL-17, and TNF-α, which promote osteoclastogenesis and skew bone homeostasis toward pathological bone resorption[31–34]. Interestingly, our previous study[24] showed that HIF-1α and HIF-2α were stabilized under normoxia in cells cultured without any cytokines in osteogenic differentiation medium and in trabecular bone tissues during osteoporotic bone loss. It cannot be entirely ruled out that the stabilization of HIF-1α under normoxic conditions, as posited in our current research, may be influenced by locally occurring hypoxic conditions in in vivo pathological settings. The vascularized bone microenvironment, influenced by systemic circulation, is not hypoxic compared to various other organs[35,36]. Kim et al. directly measured in vivo oxygen levels in the bone architecture of rats, revealing that vascularized bone is a high-$O_2$ area compared to vessel-free cartilage with low-$O_2$ levels and bone marrow with medium-$O_2$ levels[37]. They suggested that the higher expression of HIF-1α in high-$O_2$ areas is associated with ROS signaling. In our study, we conducted all experiments under normoxia and observed stabilization of HIF-1α and

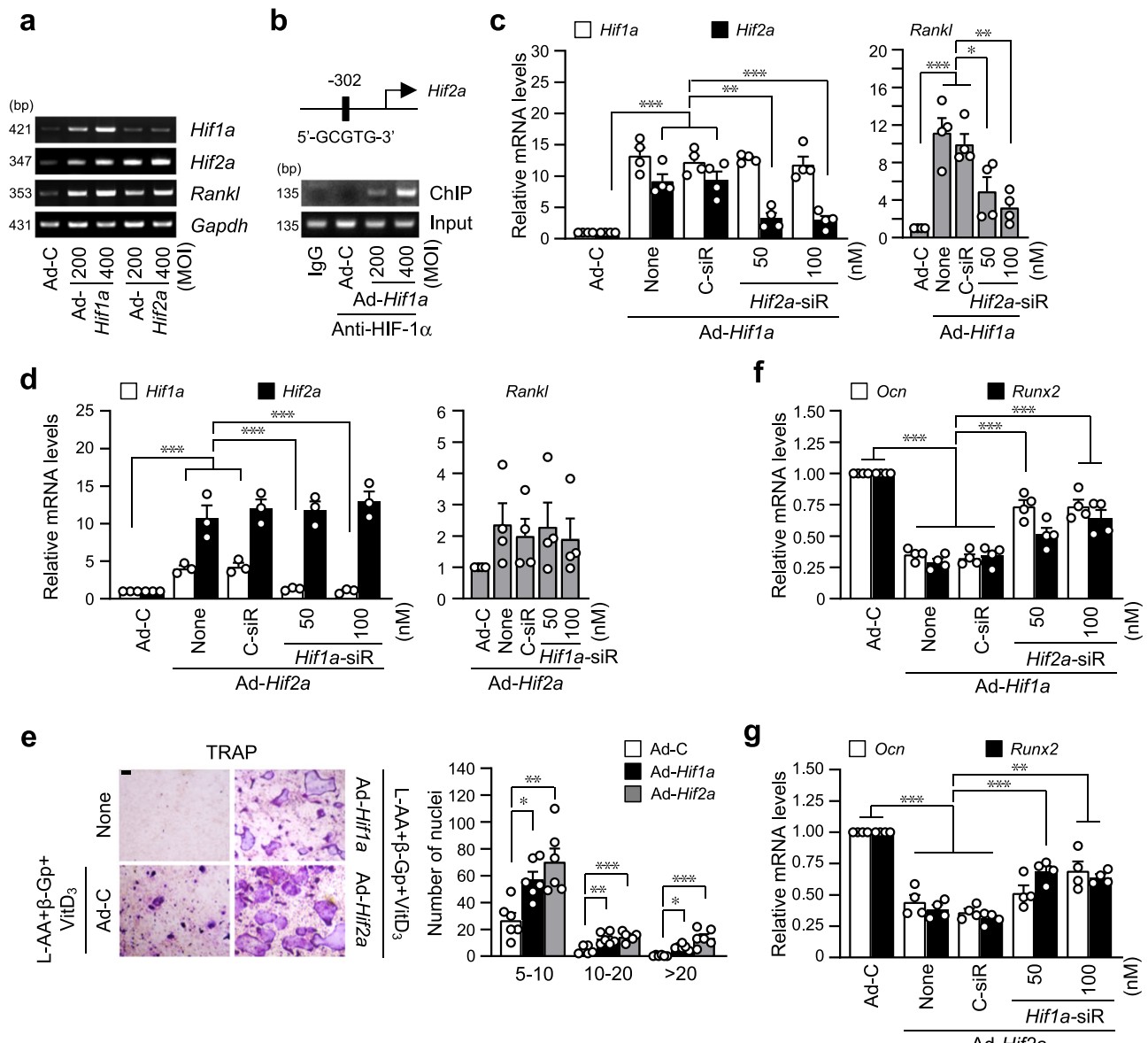

**Fig. 5 | HIF-1α indirectly promotes RANKL-mediated osteoclastogenesis via HIF-2α. a** Primary cells were infected with the indicated MOI of Ad-*Hif1a* or Ad-*Hif2a* on the 3rd day of differentiation and cultured until day 6. Expression pattern of *Hif1a*, *Hif2a*, and *Rankl* was determined using RT-PCR, and representative data were shown (*n* = 3). **b** ChIP was performed with anti-HIF-1α antibody and primers spanning the HRE motif of the promoter region of the *Hif2a* gene (*n* = 3). **c, d** qRT-PCR analyses of *Hif1a, Hif2a*, and *Rankl* in pre-osteoblasts transfected with *Hif2a* siRNA in *Hif1a*-overexpressing cells (**c**) and transfected with *Hif1a* siRNA in *Hif2a*-overexpressing cells (**d**) (*n* ≥ 3). **e** Calvarial pre-osteoblasts infected with Ad-C, *Hif1a*, or *Hif2a* adenovirus were cultured with BMMs in a medium containing L-AA (50 μg/ml), β-Gp (5 mM), and VitD₃ (10 nM) for 5 days. TRAP staining and quantitative analysis of multinucleated cells are shown (*n* = 6). Scale bar, 100 μm. **f, g** qRT-PCR analyses of *Ocn* and *Runx2* in pre-osteoblasts transfected with *Hif2a* siRNA in HIF-1α-overexpressing cells (**f**) and transfected with *Hif1a* siRNA in HIF-2α-overexpressing cells (**g**) (*n* = 4). Values are presented as the mean ± SEM. *$P < 0.05$, **$P < 0.01$, and ***$P < 0.001$.

HIF-2α during osteoblast differentiation of pre-osteoblasts under these conditions. We hypothesized that certain factors in the normal physiological or pathological environment could induce stabilization of the HIF-α subunit under normoxia. We investigated the effects of osteogenic differentiation-induced normoxic stabilization of HIF-1α in osteoblasts on regulating osteoblast differentiation and osteoblast-mediated osteoclast activation. Although we did not specifically investigate the signaling pathway of HIF stabilization during osteoblast differentiation under normoxia in the current study, we recognize the importance of clarifying this aspect in further studies.

In this study, we determined similar but slightly different regulation by HIF-1α compared to HIF-2α in regulating bone remodeling. The RNA-seq analysis to determine gene expression profile by HIF-1α and HIF-2α overexpression during osteoblast differentiation (Fig. 1) showed that about 62.7% DEGs of a total of 40,213 genes were similarly regulated, and about 10.9% and 26.4% DEGs were differentially regulated by HIF-1α and HIF-2α, respectively (Fig. 1d). Performing additional experiments comparing the RNA-seq results from osteoblasts with knockdown of each HIF isoform would provide more robust evidence for conclusive insights. In the current study, HIF-1α and HIF-2α expression increased continuously until about the 15-day, starting from the 3-day in vitro culture with osteogenic differentiation medium before osteoblast marker gene, OCN expression (Supplementary Fig. 1, Supplementary Fig. 4). We observed that overexpression of HIF-1α delayed BMP-2-induced bone regeneration by inhibiting osteogenic differentiation (Fig. 2c), and HIF-1α overexpression inhibited the expression and activity of RUNX2, which resulted in the inhibition of *Ocn* expression, a transcriptional target of RUNX2. Regarding the studies showing that hypoxia signaling inhibits osteogenic differentiation of

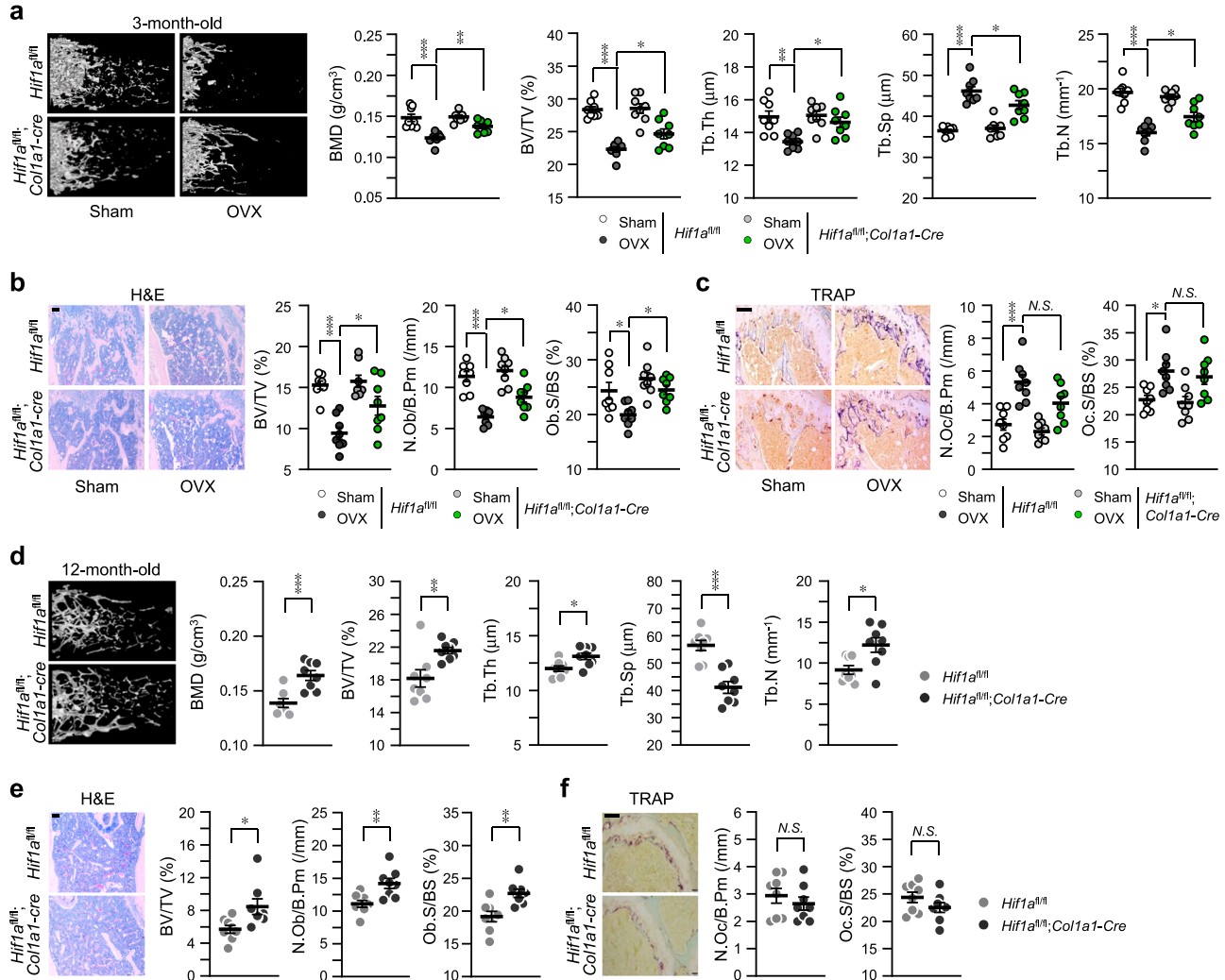

**Fig. 6 | Osteoblast-specific deprivation of *Hif1a* alleviates osteoporotic bone loss.**
**a–c** Quantitative μCT analysis of femoral trabecular bones in OVX- or sham-operated 3-month-old *Hif1a*^fl/fl and *Hif1a*^fl/fl;*Col1a1-Cre* mice (*n* = 8). **d–f** μCT analysis in 12-month-old *Hif1a*^fl/fl and *Hif1a*^fl/fl;*Col1a1-Cre* mice (*n* = 8). Representative images of μCT reconstructions (**a, d**), H&E staining, Scale bar, 100 μm (**b, e**), and TRAP staining, Scale bar, 100 μm (**c, f**). Bone-forming parameters, such as BMD, BV/TV, Tb.Th, Tb.Sp, Tb.N, N.Ob/B.Pm, and Ob.S/BS, were assessed using μCT measurements (**a, d**), bone histomorphometric analysis (**b, e**), and bone resorption parameters, N.Oc/B.Pm and Oc.S/BS were obtained from TRAP staining of the metaphyseal regions of femurs (**c, f**). Values are presented as means ± SEM (N.S. not significant, *$P < 0.05$; **$P < 0.01$, and ***$P < 0.001$). The effects of OVX and genotypes (*Hif1a*^fl/fl, *Hif1a*^fl/fl;*Col1a1-Cre*) and their interaction were tested using two-way ANOVA (**a**, BMD: interaction = 0.0404, OVX < 0.0001, genotype = 0.0162).

mesenchymal stem cells by suppressing RUNX2 via transcriptional repression of *Twist2*[38] and HIF-2α directly binds to the promoter of *Twist2* gene[23], we found that HIF-1α was also a transcriptional activator of *Twist2* gene. Thus, HIF-1α inhibited osteoblast differentiation by regulating the TWIST2-RUNX2-OCN axis, in a similar way of HIF-2α.

In our previous report[23], HIF-2α expression was increased during in vitro osteoclast differentiation of BMMs, whereas no significant differences in HIF-1α expression were observed. We found that overexpression of HIF-2α resulted in increased osteoclast formation and the number of multinucleated giant cells[23]. However, overexpression of HIF-1α did not lead to any changes in osteoclast formation and resorption. The effects of HIF-1α on osteoclast formation and activation have been controversial. Some studies have observed an increase in osteoclast differentiation following HIF-1α stimulation[39–41], while others have noted a decrease[42,43]. Shirakura et al.[44] and Wang et al.[45] suggested the possibility that the increase in osteoclast differentiation under hypoxia is the result of crosstalk between osteoblasts and osteoclasts. HIF-1α exhibited a slightly different signaling pathway on the crosstalk between osteoblasts and osteoclasts. The ratio of RANKL and OPG expressed by osteoblasts is the key regulator for the interplay between osteoblasts and osteoclasts in bone homeostasis[46].

RANKL expressed by osteoblasts binds to its receptor RANK on osteoclast precursors and induces osteoclast activation[47]. OPG produced by osteoblasts acts as a soluble decoy receptor for RANKL and modulates the balance between osteoblasts and osteoclasts[8,48]. *Hif1a* overexpression increased the ratio of *Rankl* to *Opg* expression by osteoblasts (Fig. 4a–d), but, unexpectedly, osteoblast-specific depletion of HIF-1α did not affect osteoclast activation (Fig. 4f). We also found that the knockdown of *Hif1a* with siRNA did not block *Rankl* expression, although HIF-1α remarkably upregulated RANKL expression. Our previous study showed that HIF-2α directly promotes *Rankl* transcription in osteoblasts[23]. Furthermore, HIF-2α was identified as one of the genes upregulated by *Hif1a* overexpression from the data of RNA-seq analysis (Fig. 1f). Thus, we thought that HIF-1α could be a direct transcriptional activator of *Hif2a* and it was verified by the results of the ChIP assay and reciprocal overexpression and knockdown experiments as shown in Fig. 5. The knockdown of *Hif2a* significantly blocked the *Hif1a* overexpression-mediated increase of *Rankl* expression, whereas the knockdown of *Hif1a* showed no effect on increased *Rankl* expression by *Hif2a* overexpression. The co-culture experiment showed osteoclast activation by HIF-1α-HIF-2α axis-mediated RANKL expression. Osteoclast activation of BMMs co-cultured with *Hif1a* overexpressed osteoblasts was

lesser than those with *Hif2a* overexpressed cells, which indicates that HIF-1α is an upstream effector of HIF-2α, a critical factor for RANKL-mediated osteoclastogenesis[23]. The doubt that HIF-1α-mediated inhibition of osteoblast differentiation might be also mediated by HIF-2α was eliminated by the data of Fig. 5f, g. Through the reciprocal overexpression and knockdown experiments, the knockdown of *Hif2a* significantly restored *Hif1a* overexpression-mediated suppression of *Runx2* and *Ocn* expression, and the pair of *Hif1a* siRNA and Ad-*Hif2a* also showed the same restored effect. Thus, HIF-1α and HIF-2α exhibited the same effect on the inhibition of osteoblast differentiation by sharing the TWIST2-RUNX2-OCN axis, but it was independent regulation by HIF-1α and HIF-2α. However, HIF-1α showed slightly different signaling pathways in the osteoblast-mediated osteoclastogenesis, but it was dependent on HIF-2α.

The results obtained in this study yielded somewhat challenging interpretations, among which is the difference in bone mass as depicted in Figs. 3 and 6. 4-month-old male *Hif1a*fl/fl;*Col1a1-Cre* mice exhibited an increase in bone mass (Fig. 3), whereas 3-month-old female sham groups of *Hif1a*fl/fl and *Hif1a*fl/fl;*Col1a1-Cre* mice in OVX experiments showed no discrepancy in bone mass (Fig. 6a, b). One possible explanation for this inconsistency is gender-dependent differences in responses associated with specific sex steroids and growth hormone levels[49–51]. However, further studies are required to clarify this aspect. Additionally, in our study, bone analyses were conducted using osteoblast-specific *Hif-1α* knockout mice, revealing different findings compared to those of previous studies. In our study, depletion of osteoblast-specific *Hif-1α* in mice resulted in a significant increase in bone mass through modulation of osteoblast functions. This inconsistency could be explained in two ways. Firstly, we utilized a different *Cre* transgenic model, *Col1a1-Cre*, for generating osteoblast-specific conditional knockout mice, in contrast to the *Ocn-Cre* transgenic mice used by previous studies. *Col1a1* is known to be expressed earlier than *Ocn* during osteogenesis. To further examine the expression patterns of osteogenic markers and HIF-1α, we assessed *Hif-1α*, *Col1a1*, and *Ocn* levels during osteogenesis in human mesenchymal stem cells. Consistent with the results obtained using pre-osteoblast cells (Supplementary Fig. 1a), the mRNA expression pattern of *Hif-1α* closely resembled that of *Col1a1* but decreased at the *Ocn*-expressing stage. In addition, conflicting results were reported by Wu et al.[52] They suggest that HIF-1α is necessary within the context of combined PHD2 and PHD3 inactivation to inhibit bone resorption through an osteoblastic mechanism of OPG. Inconsistent results on the direct regulation of *Opg* by HIF-1α were presented in our current study (Fig. 4a). In their report, *Osterix (Osx)-Cre* mice were used for generation of osteoblast-specific conditional KO mice. The mouse *Osx-Cre* transgenic line expresses the *Cre* recombinase in committed osteoblast progenitors in both endochondral and membranous-derived bones[2]. Immature osteoblasts can be targeted by using the *Osx-Cre* transgenic mice, while mature osteoblasts can be targeted by using the *Col1a1-Cre* transgenic mice (especially the 2.3 kb *Col1a1-Cre* mice used in our experiment) based on their rich expression of type I collagen, the main constituent of bones[53]. This finding supports the differing phenotypes observed between our study and earlier ones. Another possible explanation for the discrepant results could be the variations in the ages of mice used for experiments. It has been suggested the HIF may exert differential effects depending on age and bone cycle (modeling vs. remodeling) in our previous study[23]. While previous studies analyzed juvenile or young adult mice, we utilized mature mice to assess the regulatory role of HIF-1α in bone remodeling and osteoporotic bone loss. These hypotheses should be further investigated using various *Cre* mouse lines and multiple time points (ages) in future studies.

The ultimate question of our study was attributed to the similar regulation in osteoblast differentiation but different regulation in osteoblast-mediated osteoclast activation. Both HIF-1α and HIF-2α expression was increased during osteoblast differentiation, and their effects on osteoblast differentiation were the same; therefore, both HIF-1α and HIF-2α expression during osteoblast differentiation seemed to be redundant. However, we found their different effects on osteoclast activation. In the results of in vivo experiments using OVX- or aging-induced osteoporosis model, osteoblast-

specific knockout of HIF-1α (*Hif1a*fl/fl;*Col1a1-Cre* mice) showed no differences in osteoclast differentiation and activation compared to control, *Hif1a*fl/fl mice, although *Hif1a* cKO did show a significant increase in BMD and bone parameters (Fig. 6). In a previous report[23], HIF-2α expression was increased during osteoclastogenesis of BMM, but not HIF-1α, and HIF-2α specifically promoted osteoclastogenesis by directly promoting *Traf6* and RANKL-mediated osteoclastogenesis. This finding indicates that HIF-2α has more catabolic function in regulating bone remodeling than HIF-1α.

In conclusion, in the normal physiology of bone remodeling, HIF-1α could fine-tune osteoblast-mediated osteoclastogenesis by regulating HIF-2α expression, enabling the maintenance of bone homeostasis. Especially in the bone pathophysiologic microenvironment, some inflammatory cytokines may enhance the normoxic stabilization of HIF-1α and/or HIF-2α and skew the equilibrium of bone homeostasis toward bone resorption because HIF-2α promotes osteoclastogenesis and RANKL-mediated osteoclast activation and HIF-1α enhances HIF-2α function. We suggest that the normoxic stabilization of HIF-1α and HIF-2α during osteoblast differentiation is essential to osteoporotic bone loss. Especially in bone pathophysiology, several factors, including inflammatory cytokines, have the potential to stimulate the stabilization of HIF-1α and HIF-2α, which contributes to disturbances in bone homeostasis. In this study, although we didn't investigate what factors induce normoxic stabilization of HIF-1α and HIF-2α and it should be further studied, we can emphasize the need to block both HIF-1α and HIF-2α to promote bone formation and regeneration. However, it is more necessary to inhibit HIF-2α to delay or inhibit osteoporotic bone loss. This study provides therapeutic advances in other human diseases correlated with bone loss.

## Methods
### Ethics statement and experimental mice
*Hif1a*fl/fl mice were obtained from Jackson Laboratory (#007561, Sacramento, CA, USA), and *Col1a1-Cre* mice were kindly donated by Dr. Je-Yong Choi (Kyungpook National University, Daegu, Korea). To establish osteoblast-specific *Hif1a* knockout mice, *Hif1a*fl/fl mice were backcrossed against *Col1a1-Cre* mice. Critical-sized (5-mm diameter) defects were created in 6-week-old C57BL/6 J male mice for calvarial defect models as previously described in ref. 23. 300 ng bone morphogenetic protein (BMP)-2 containing collagen sponges were applied to cover the defects. After 2 weeks, calvarial defect size was measured and further analyzed. Male mice were used for all experiments except for the ovariectomized (OVX) osteoporosis models. For the OVX models, a 5-mm dorsal incision and sham operation (as a control) were performed using 8-week-old female mice. After four weeks, the OVX mice were sacrificed for further analysis. All procedures for animal care and experiments were approved by the Institutional Animal Care and Use Committee (IACUC) of Chonnam National University (Gwangju, Korea).

### Primary culture of pre-osteoblasts and osteogenic differentiation
All experiments conducted in vitro culture in our study were under normoxic conditions. In general cell culture models, normoxic conditions are maintained in standard humidified cell culture incubators with 5% $CO_2$, where the oxygen concentration is approximately 18% $O_2$ (v/v), equivalent to an oxygen tension or partial pressure ($pO_2$) of 138 mmHg[54,55]. Primary culture of calvarial pre-osteoblasts and induction of osteoblast differentiation were performed as previously described in refs. 23,24. In brief, calvarial bones were isolated from 3-day-old pups of mice and enzymatically digested with dulbecco's modified eagle's medium (DMEM) containing 0.1% type II collagenase (Sigma–Aldrich, St. Louis, MO, USA) and 0.25% trypsin/EDTA at 37 °C for 20 min. The isolated pre-osteoblasts were cultured in complete media, DMEM (GIBCO, Grand Island, NY, USA) containing 10% fetal bovine serum and 1% penicillin/streptomycin. After 3 days, $1 \times 10^5$ cells were plated in a 35-mm culture dish and cultured in osteogenic differentiation media (DM) containing 50 μg/ml L-ascorbic acid (L-AA, A0278, Sigma–Aldrich) and 5 mM β-glycerophosphate (β-Gp, sc-220452A Chemcruz, Dallas, TX, USA). Adenoviral infection was performed on day

3 at the indicated MOI (Multiplicity of infection). For siRNA-mediated knockdown, primary cells were transfected with *Hif1a* siRNA, *Hif2a* siRNA, or *Twist2*-siRNA (Dharmacon, La Fayette, CO, USA) on differentiation day 3 using Lipofectamine RNAiMAX (Invitrogen, Carlsbad, CA, USA) following the manufacturer's recommendations. Non-targeting siRNA (scrambled; Bioneer, Daejeon, Korea) was used as a negative control. After gene overexpression and knockdown, cells were harvested on 6 days of osteoblast differentiation for further experiment.

## Co-culture of osteoblasts and bone marrow-derived macrophages (BMMs)

To perform the co-culture of osteoblasts and BMMs, primary calvarial pre-osteoblasts and BMMs were prepared as follows: Bone marrow was isolated from the long bones of 6- to 8-week-old mice and flushed with serum free α-MEM. Bone marrow cells were cultured in complete α-MEM for 24 h, after which non-adherent cells were collected and cultured in complete α-MEM in the presence of 30 ng/ml of M-CSF (Peprotech, Rocky Hill, NJ, USA) for 3 days. Adherent BMMs ($2 \times 10^4$ cells per well in a 48-well plate) were then maintained in complete α-MEM containing 30 ng/ml of M-CSF for 24 h. Subsequently, they were co-cultured with primary calvarial pre-osteoblasts ($4 \times 10^3$ cells/well) infected with Ad-C, Ad-*Hif1a*, or Ad-*Hif2a* in the presence of 100 ng/ml BMP-2, 50 μg/ml L-AA, 5 mM β-Gp, and 10 nM 1,25-dihydroxyvitamin $D_3$ (VitD$_3$) for 5 days[23]. Cells were fixed and applied for tartrate-resistant acid phosphate (TRAP) staining, and images were obtained using LAS (Leica ApplicationSuite) V4.1 program (Leica, Switzerland).

## RNA sequencing (RNA-seq) analysis

For RNA-seq analysis, RNA samples were prepared from primary cultured calvarial pre-osteoblasts infected with control adenovirus (Ad-C), Ad-*Hif1a*, or Ad-*Hif2a*. Adenoviral infection was performed on day 3, followed by culturing the cells in osteogenic DM until day 6. RNA-seq service was provided by Ebiogen (Seoul, Korea). Gene expression profiling and graphic visualization were performed using the ExDEGA tool and ExDEGA Graphic Plus software. For the analysis of differentially expressed genes (DEGs), significant gene selection was filtered using an absolute value of fold change (FC) > 2, normalized data (log2) > 4 (Fig. 1d, e) or normalized data (log2) > 2 (Fig. 1f), and *p*-value (paired *t*-test) < 0.05 as the cut-off.

## Reverse transcription-polymerase chain reaction (RT-PCR) and quantitative real-time (qRT)-PCR

Total RNA was isolated using TRI reagent (TR118, MRC, Cincinnati, OH, USA), and cDNA was obtained with a reverse transcription kit (TOPscript RT DryMIX, Enzynomics, Daejeon, Korea). The AmpOne™ Tap DNA Polymerase Mix (GeneAll, Seoul, Korea) and Applied Biosystems and SYBR premix Ex Taq (RR420, TaKaRa, Japan) were used for conventional PCR and qRT-PCR, respectively. All primer pairs are for the mouse genes and the sequence information of the primers and PCR conditions are listed in Supplementary Table 1. The individual transcript levels of each target gene were normalized with *Gapdh*, and relative levels were represented as a fold change relative to the indicated controls.

## Western blotting

Lysis buffer (RIPA) for western blotting was prepared using 50 mM Tris-HCl (pH 8.0), 150 mM NaCl, 5 mM NaF, 1% NP-40, 0.2% SDS, 0.5% deoxycholate, a protease inhibitor cocktail, and a phosphatase inhibitor cocktail (Roche, Basel, Switzerland). Protein samples were applied for SDS-PAGE and transferred to nitrocellulose membranes. After blocking the membrane with 5% skim milk, the samples were incubated with indicated primary antibodies at 4 °C overnight. The following antibodies were used: anti-HIF-1α (NB100-134; Novus biologicals, Centennial, CO, USA), anti-HIF-2α (NB100-122; Novus biologicals), anti-Lamin A/C (2032 S; Cell signaling, Davnvers, MA, USA), anti-OCN (AB10911, Merck, Darmstadt, Germany), and anti-β-Actin (A3584, Sigma–Aldrich). Protein levels were detected using horseradish peroxidase-conjugated secondary antibodies and an ECL detection system (RPN2235, Cytiva, Buckinghamshire, UK).

## Enzyme-linked immunosorbent assay (ELISA)

Secreted RANKL protein levels in the culture medium were measured using a RANKL-ELISA kit (ab-100749, Abcam, Cambridge, UK), following the manufacturer's instructions.

## Immunocytochemistry

Cells cultured on 12-mm coverslips were fixed with 3.5% paraformaldehyde, permeabilized with 0.1% Triton X-100, and blocked with 1% BSA. Then, cells were incubated with primary antibodies, anti-HIF-1α, anti-HIF-2α or anti-RANKL antibodies, and an Alexa-594-conjugated secondary antibody. DAPI (4′,6-diamidino-2-phenylindole) was used for staining nuclei. Fluorescence images were captured by Zeiss microscope and analyzed by IMT isolution FL/Auto software. Positive staining cells were counted using the NIH ImageJ program (version 1.47, National Institutes of Health, Bethesda, MD, USA).

## Chromatin immunoprecipitation (ChIP) assay

ChIP assays were performed as described previously[24]. In brief, crosslinks of genomic DNA and proteins were achieved by adding 1% formaldehyde to the cell culture medium at room temperature for 10 min. Cells were lysed with RIPA buffer and then sonicated to cause fragmentation of DNA. The samples were incubated with 2 μg of anti-HIF-1α antibody or anti-IgG at 4 °C overnight. The precipitated complex of antibody and chromatin fragments were applied for PCR with specific primers for *Twist2* or *Hif2a* promoter. The primer sequences for the ChIP assay are listed in Supplementary Table 1.

## Luciferase reporter assay

Primary calvarial pre-osteoblasts infected with Ad-*Hif1a* were transfected with pGL3-6xOSE or pGL3-OG2, RUNX2-responsive luciferase reporter constructs, plus a CMV-β-galactosidase construct as an internal control[56]. Cell lysates were used for measuring luciferase reporter activity using a luciferase reporter assay system (Promega, Madison, WI, USA). The individual values were normalized with those of β-galactosidase activity.

## Alkaline phosphatase (ALP) and alizarin red-S (ARS) staining

For ALP staining, pre-osteoblasts were cultured for 7 days in DM media. After fixation with 4% formaldehyde, cells were rinsed with deionized water and stained with 5-bromo-4-chloro-3-indolyl phosphate (BCIP®)/nitro blue tetrazolium (NBT) Liquid Substrate solution (Sigma–Aldrich) for 30 min in a dark room. For ARS staining, cells were cultured for 14 days in DM media, fixed with 10% formalin for 15 min, and stained with 2% Alizarin red-S solution (ARS, Sigma–Aldrich) for 45 min at room temperature[57].

## X-ray microcomputed tomography (μCT) and bone histomorphometry analysis

Mouse femurs were fixed in 10% neutral buffered formalin and analyzed with high-resolution μCT (Skyscan 1172 system, Bruker, Aartselaar, Belgium) as described previously[23]. Image reconstruction software (NRecon; Bruker) was used with identical thresholds for all samples (0–6000 in Hounsfield units) for reconstructing serial cross-section images. For the trabecular bones in the proximal femurs, a region of interest comprising 300 total steps starting at 30 steps away from the growth plate was manually designated for the trabecular bones in the proximal femurs. Femoral morphometric parameters, such as bone mineral density (BMD), bone volume per tissue volume (BV/TV), trabecular thickness (Tb.Th), trabecular separation (Tb.Sp), and trabecular number (Tb.N), were determined with CTAn data analysis software. Quantitative histomorphometric parameters, the number of osteoblasts per bone perimeter (N.Ob/B.Pm), and the osteoblasts surface per bone surface (Ob.S/BS) were analyzed using the data of H&E staining with OsteoMeasure software (Osteometrics, Inc., Decatur, GA, USA). For osteoclast parameters, the number of osteoclasts per bone perimeter (N.Oc/B.Pm) and osteoclasts surface per bone surface (Oc.S/BS) were analyzed with TRAP staining data.

## Immunohistochemistry

Isolated distal femurs were fixed with 10% neutral buffered formalin and decalcified in 0.5 M EDTA (pH 7.4). After serial dehydration procedures, bone tissues were embedded in paraffin. Sectioned slices at 5-μm thickness on slide glasses were incubated in 3% $H_2O_2$ to block endogenous peroxidase activity and treated with 0.1% trypsin for 40 min at 37 °C for antigen retrieval. After blocking with 1% BSA for 30 min, the sectioned tissues were incubated with anti-HIF-1α. EnVision-HRP (K5007, Dako, Denmark) and AEC substrate kit (SK-4200, Vector laboratories, Newark, CA, USA) were used for visualization, and hematoxylin (Dako) was used for counterstain.

## Statistical analysis

All experiments were repeated at least three times. Results were reported as the mean ± SEM. μCT and histomorphometric parameters were presented in bar charts with scatter plots. All statistical analyses were performed using GraphPad Prism version 7 (GraphPad Software, Inc., San Diego, CA, USA). All quantified data were first tested for conformation to a normal distribution using the Shapiro–Wilk test, followed by analysis with two-tailed Student's $t$-test for quantitative variables between the means of two groups or one-way ANOVA or two-way ANOVA followed by Tukey's post hoc tests (multi-comparison) for variables between the means of three or more independent groups, as appropriate. The $n$-value was the number of independent experiments or mice. Results were considered statistically significant when $p$-values were less than 0.05.

## Reporting summary

Further information on research design is available in the Nature Portfolio Reporting Summary linked to this article.

## Data availability

RNA-seq data was obtained from the Gene Expression Omnibus database under accession number GSE271730. Source data for the graphs and charts are provided as Supplementary Data 1. All data that support the findings of this study are available from the corresponding author upon reasonable request.

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

## Acknowledgements

This work was supported by National Research Foundation of Korea (NRF) grants funded by the Korean government (MSIT) (2019R1A5A2027521, 2021R1A2C3005727, 2021R1F1A1062446, 2022R1C1C2013357), and a Korean Fund for Regenerative Medicine (KFRM) grant funded by the Korean government (the Ministry of Science and ICT, the Ministry of Health & Welfare) (22A0104L1).

## Author contributions

J.-H.R. and Y.H.H. conceived and designed the study. S.Y.L. and S.-J.K. performed the in vitro and in vivo experiments and analyzed data. S.-J.K. prepared cells for RNA-seq. S.Y.L. performed computational analysis for RNA-seq. K.H.P., G.L., and Y.O. helped with the data analysis. Y.H.H. wrote the manuscript. All authors revised and agreed on the final version of the manuscript.

## Competing interests

The authors declare no competing interests.
