## [Peer Review File · Communications Biology]

Reviewers' comments:

Reviewer #1 (Remarks to the Author):

Thank you for the opportunity to review this manuscript by Lee et al. The authors report that in vitro over-expression of HIF1a has an inhibitory effect on osteoblast differentiation and increases RANKL expression, despite a limited effect on osteoclast formation. HIF1a over-expression also increases HIF2a expression. In vitro silencing of HIF2a blocked HIF1a over-expression-induced RANKL expression, indicating that HIF-1a induced RANKL expression is mediated by HIF-2a. These data indicate that HIF-1a acts upstream of HIF-2a in the regulation of RANKL-mediated osteoclastogenesis.

Conversely, in vitro HIF-1a-siRNA and in vivo knock-down of HIF1a in pre-osteoblasts (HIF-1a-Col1a1-Cre mouse) improves bone formation parameters with a limited effect on osteoclasts. HIF-1a-Col1a1-Cre-ovariectomised mice also exhibited slightly improved bone parameters relative to OVX-fl/fl mice, but not sham mice.

Overall, I believe that the conclusions drawn are generally supported by the presented data. To my knowledge, the conducted statistical analyses appear to be appropriate and valid. The below questions, comments and recommendations could clarify some aspects I feel are currently missing and unclear in the manuscript:

1. What is the evidence you have (both in vivo and in vitro) that the model really is normoxic stabilisation and not just, say, poor oxygen diffusion in the culture medium or (local) inflammation due to OVX/ ageing etc. causing hypoxia?
2. In Figure 3, μ CT BV/TV data are shown twice (i.e., Figs. C & D). What are the differences? If there are differences, these need to be clarified better, please.
3. Figures 2 & 6: In the OVX HIF1a knockdown model, what is the effect on HIF2 and Twist2 expression? What about RANKL expression and secretion?
4. Figures 4 & 5: what effect does HIF1a over-expression have on osteoclast formation and resorption in vitro? Could you provide the quantitative data alongside the qualitative Figure 5e, please?
5. Figure 6a: When comparing Sham mice between fl/fl and Col1a1-Cre mice (white vs. light grey dots), the increase in BMD, BV/TV, Tb.Th etc. between HIF1a knockdown reported in Fig. 3a are no longer seen. I note that mice in Fig.3 are 4-months and mice used in Fig. 6 are 3 months old. Is this the main explanation for this discrepancy?
6. RNAseq methods & results section: please include the the day/ timepoint that the sample was taken for RNAseq. Currently, this information is only provided in Fig.1 legend.

7. Methods line 360: Are the bone marrow macrophages freshly isolated and immediately cultured with primary osteoblasts?
8. Discussion: I feel that there could be better contextualisation of the reported results to the wider literature. As one example, the relationship between the authors' previously published work on HIF2a could be included to link the current interplay between HIF1a and HIF2a, particularly in the context of osteoclast formation.

Reviewer #2 (Remarks to the Author):

This manuscript describes the effects of manipulation of HIF isoforms on osteoblastogenesis and osteoclastogenesis. The effects of Cre driven HIF deletion in osteoblasts on skeletal development is also assessed. Overall the manuscript describes a comprehensive set of experiments. However, especially in relation to in vivo experiments on transgenic mice, the effects of skeletal phenotype described are the opposite of what has been previously described in the literature. For example, the authors describe how deletion of HIF-1a using a Col1a1 driver increases trabecular bone mass. Published studies using *Osx-cre* demonstrated reduced bone mass (PMCID: PMC4403258), *Bglap-cre* ; decreased bone volume (PMCID: PMC1878533 and PMID: 19899108) and *DMP-1 cre* no effect on bone volume (PMID: 30172741). Importantly, the Cre drivers used in these studies target deletion of HIF1a in osteoblastic cell lineage cells both earlier and later than the Col1a1 driver.

The same is true regarding OVX studies in Figure 6. In this study deletion of HIF 1a reduced the amount of bone loss induced by OVX. However, published literature shows that HIF 1 expression decreases with OVX (murine studies), and that *Vhl* deletion which upregulates HIF1 expression protects the skeleton from OVX-induced bone loss (PMID: 22193550).

Further – numerous studies using a variety of Cre-drivers demonstrate dramatically increased bone mass in mice with deletion of *Vhl* which upregulates expression/stabilization of both HIF isoforms. These published studies are not discussed and why the current study is in direct opposition is not addressed.

Specific comments:

Figure 1: Data showing expression levels of HIFs at day 6, in figure 1a is not the same as that shown at the same timepoint in the supplementary figure 1. HIF1 and HIF2 are similar in Fig 1 but the supplementary figure suggests HIF2 expression exceeds HIF1.

Panel b shows immunostaining for HIF1 –similar data should be provided for HIF2.

Define normoxia in the context of your studies.

Figure 2

Please clarify at what differentiation day target mRNAs were measured – text says cells were infected with adenovirus at day 3.

The data is somewhat over interpreted. It is stated that Twist 2 knockdown restored the decrease of Ocn and Runx2 – this was a partial effect.

Figure 3.

Please clarify where the bone images in panel a are taken from – provide a lower mag H&E image for localization.

Figure 5. Text states that Hif1a expression is not affected by Hif2a overexpression – BUT panel d shows an upregulation of both Hif1 and Hif2 with Ad-hif2a.

For panel e were osteoblasts transduced with Ad-Hif2a prior to co culture.

The strong induction of osteoclastogenesis in 5d suggests HIF 2 is the driver.

General comments

The size of the circles indicating individual data points are too large and obscure the bars and individual points themselves.

There is a lack of detail in many of the figure legends. In many cases it is unclear how long cells were cultured in osteogenic conditions. Further, the authors should provide data to show how quickly Ad-HIF transfection increases HIF1 or HIF2 and how long the effects are sustained. The same criticism can be leveled at siRNA approaches. In many experiments, cells are cultured for extended periods of time and it is unknown whether effects on HIFs are sustained.

Please check all references for applicability and that they have been cited correctly. For example, there is no mention of hypoxia or HIFs in some of these references “Remarkably, HIF- α can be stabilized even in normoxia by inflammatory factors, such as interleukin (IL)-70 1 β , IL-6, IL-17, and tumor necrosis factor (TNF)- α , in a pathophysiologic microenvironment, and normoxic stabilization of HIF-1 α induces osteoclastogenesis and pathological bone resorption^{11–16}.”

Further this interpretation of this reference is incorrect “ Loss of HIF-1a is responsible for the increased accumulation of bone matrix” line 81.

This manuscript should be thoroughly checked for grammar. In many cases grammatical errors impact the clarity of the message. The following sentence in the abstract is one such example “ Our findings conclude that HIF-1 α plays a crucial role in regulating bone homeostasis, such as

osteoblast differentiation independently on HIF-2 α and osteoclast activation via crosstalk with osteoblasts depending on HIF-2 α .”

Point-by-point responses to reviewers' comments

Comments from the reviewer #1:

Thank you for the opportunity to review this manuscript by Lee et al. The authors report that in vitro over-expression of HIF1a has an inhibitory effect on osteoblast differentiation and increases RANKL expression, despite a limited effect on osteoclast formation. HIF1a over-expression also increases HIF2a expression. In vitro silencing of HIF2a blocked HIF1a over-expression-induced RANKL expression, indicating that HIF-1a induced RANKL expression is mediated by HIF-2a. These data indicate that HIF-1a acts upstream of HIF-2a in the regulation of RANKL-mediated osteoclastogenesis.

Conversely, in vitro HIF-1a-siRNA and in vivo knock-down of HIF1a in pre-osteoblasts (HIF-1a-Coll1a1-Cre mouse) improves bone formation parameters with a limited effect on osteoclasts. HIF-1a-Coll1a1-Cre-ovariectomised mice also exhibited slightly improved bone parameters relative to OVX-fl/fl mice, but not sham mice.

Overall, I believe that the conclusions drawn are generally supported by the presented data. To my knowledge, the conducted statistical analyses appear to be appropriate and valid. The below questions, comments and recommendations could clarify some aspects I feel are currently missing and unclear in the manuscript:

Response:

We found the reviewer's comments to be constructive and helpful in revising our manuscript, which has been amended to include additional experiments and extensive related discussion, as recommended. We sincerely believe that our manuscript has been significantly improved as a result.

Specific comments

Comment #1. What is the evidence you have (both *in vivo* and *in vitro*) that the model really is normoxic stabilisation and not just, say, poor oxygen diffusion in the culture medium or (local) inflammation due to OVX/ ageing etc. causing hypoxia?

Response:

We appreciate the reviewer's concerns. In this study, we determined that HIF-1 α , a well-known transcription factor stabilized under hypoxic condition, can be stabilized even under normoxic condition during osteoblast differentiation. Several studies also examined accumulation of HIF-1 α protein level and stimulation of the HIF-1 α pathway by cytokines or hypoxic mimicking agents such as cobalt or L-mimosine; gene therapy; and iron chelators¹. As concerned by the reviewer, *in vitro* and *in vivo* system we presented in this study might be affected by poor oxygen diffusion, resulted in hypoxia-induced stabilization of HIF-1 α .

in vitro In standard humidified cell culture incubators maintained at 5% CO₂, oxygen levels are equal to 18.6% O₂ (v/v), equivalent to an oxygen tension or oxygen partial pressure (pO₂) of 138 mmHg². To the best of our knowledge based on our investigation, there have been no reports indicating changes in the concentration of oxygen during osteoblast differentiation. According to a study investigating the impact of hypoxia on the bone-forming ability of osteoblasts, osteoblasts cultured in 20% oxygen for 18 to 24 days exhibited abundant bone nodule formation. However, reducing the oxygen level to 2% resulted in a decrease in the area of bone nodules³. Considering this, as osteoblast differentiation was sufficiently induced in our culture condition, it is difficult to argue that hypoxia played a role in the culture conditions promoting osteoblast differentiation in our study.

in vivo Bone is highly vascularized, but different regions of bone tissue characterized by different oxygen levels and oxygen gradient. The level of oxygen reaching bone tissue is thought to be around 6.6–8.6% O₂, as measured in bone aspirates⁴. This means that bone environment is physiologically normoxic condition and to the best of our knowledge there have been no reports indicating changes in the concentration of oxygen in pathological situations including OVX and ageing *in vivo*. However, the bone marrow is a relatively hypoxic microenvironment⁵ and hypoxia resulting from insufficient blood supply to the bone is known in osteonecrosis of the femoral head (ONFH)⁶. In addition, hypoxia clearly presents itself as a regulator of bone cell functions. Considering this, it is entirely conceivable that exposure to small changes in either inspired O₂ or O₂ delivery may influence cell homeostasis, namely by stimulation of HIF pathways^{1,7,8}. Taking into account the various possibilities mentioned above, we have incorporated additional discussion points to address these concerns.

We have modified the 'Title' and added this aspect in the 'Discussion' part in the revised manuscript.

Title (lines 1-2)

Differential but complementary roles of HIF-1 α and HIF-2 α ~~under normoxia~~ in the regulation of bone homeostasis

Discussion section (p. 11, Lines 246-249)

Interestingly, our previous study showed that HIF-1 α and HIF-2 α were stabilized under normoxia in cells cultured without any cytokines in osteogenic differentiation medium and in trabecular bone tissues during osteoporotic bone loss. It cannot be entirely ruled out that the stabilization of HIF-1 α under normoxic conditions, as posited in our current research, may be influenced by locally occurring hypoxic conditions in *in vivo* pathological settings.

[HIF-1 α stabilization under normoxic condition] In response to the comment raised by the reviewer, we have additionally provided evidence of normoxic stabilization of HIF-1 α and HIF-2 α in our studies. The figure presented below is from our previous study⁹. As depicted in this figure, both HIF-1 α and HIF-2 α were stabilized in osteoblast differentiation media (DM) under normoxic conditions. Therefore, all experiments in our study were conducted under normoxia.

[Fig. 2b] Lee, S. Y. et al. *Bone Res.* 7, 14 (2019).

In addition, Kim *et al.* showed higher HIF-1 α expression in both high- and low-O₂ areas compared to medium-O₂ areas¹⁰. According to this paper, HIF-1 α is highly expressed *in vivo* in vascularized trabecular bone, which is not a hypoxic area. This led us to hypothesize that the signaling pathway regulating osteoblast differentiation of bone marrow stem cells (BMSCs) in the bone marrow would be different from that of osteoblast differentiation of pre-osteoblasts in the trabecular bone area.

Many papers have reported that HIF-1 α can be stabilized by oncometabolites, such as lactate¹¹, succinate¹², and fumarate¹³, as well as by ROS¹⁴, even under normoxia. Kim *et al.*¹⁰ suggested that higher HIF-1 α expression in high-O₂ areas is associated with ROS signaling. In our study, we conducted all experiments under normoxia and found that HIF-1 α and HIF-2 α were stabilized under normoxic conditions during osteoblast differentiation of preosteoblasts. While we did not investigate the signaling pathway of HIF stabilization during osteoblast differentiation under normoxia, it is necessary to clarify this aspect in further studies.

We have added this aspect in the ‘Discussion’ part in the revised manuscript.

Discussion section (p. 11, Lines 249-262)

The vascularized bone microenvironment, influenced by systemic circulation, is not hypoxic compared to various other organs. Kim *et al.* directly measured *in vivo* oxygen levels in the bone architecture of rats, revealing that vascularized bone is a high-O₂ area compared to vessel-free cartilage with low-O₂ levels and bone marrow with medium-O₂ levels. They suggested that the higher expression of HIF-1 α in high-O₂ areas is associated with ROS signaling. In our study, we

conducted all experiments under normoxia and observed stabilization of HIF-1 α and HIF-2 α during osteoblast differentiation of pre-osteoblasts under these conditions. We hypothesized that certain factors in the normal physiological or pathological environment could induce stabilization of the HIF- α subunit under normoxia. We investigated the effects of osteogenic differentiation-induced normoxic stabilization of HIF-1 α in osteoblasts on regulating osteoblast differentiation and osteoblast-mediated osteoclast activation. Although we did not specifically investigate the signaling pathway of HIF stabilization during osteoblast differentiation under normoxia in the current study, we recognize the importance of clarifying this aspect in further studies.

[Reference]

1. Hannah *et al.* "Take My Bone Away?" Hypoxia and bone: A narrative review. *J Cell Physiol.* 236(2):721-740 (2021).
2. Wenger *et al.* Frequently asked questions in hypoxia research. *Hypoxia (Auckl)* 3:35-43 (2015).
3. Utting *et al.* Hypoxia inhibits the growth, differentiation and bone-forming capacity of rat osteoblasts. *Exp Cell Res.* 312(10):1693-1702 (2006).
4. Harrison *et al.* Oxygen saturation in the bone marrow of healthy volunteers. *Blood.* 99(1):394 (2002).
5. Johnson *et al.* Hypoxia and Bone metastasis disease. *Curr Osteoporos Rep.* 15(4): 231–238 (2017).
6. Yin *et al.* Effects of hypoxia environment on osteonecrosis of the femoral head in Sprague-Dawley rats. *J Bone Miner Metab.* 38(6):780-793 (2020).
7. Arnett, T. R. Acidosis, hypoxia and bone. *Arch Biochem Biophys.* 503(1):103-109 (2010).
8. Marenzana & Arnett. The Key Role of the Blood Supply to Bone. *Bone Res.* 1(3):203-215 (2013).
9. Lee *et al.* Controlling hypoxia-inducible factor-2 α is critical for maintaining bone homeostasis in mice. *Bone Res.* 7:14 (2019).
10. Kim *et al.* O2 variant chip to simulate site-specific skeletogenesis from hypoxic bone marrow. *Sci. Adv.* 9(12):eadd4210 (2023).
11. Kozlov *et al.* Lactate preconditioning promotes a HIF-1 α -mediated metabolic shift from OXPHOS to glycolysis in normal human diploid fibroblasts. *Sci Rep.* 10(1):8388 (2020).
12. Selak *et al.* Succinate links TCA cycle dysfunction to oncogenesis by inhibiting HIF-alpha prolyl hydroxylase. *Cancer Cell.* 7(1):77-85 (2005).
13. Laukka *et al.* Fumarate and Succinate Regulate Expression of Hypoxia-inducible Genes via TET Enzymes. *J Biol Chem.* 291(8):4256–4265 (2016).
14. Sasabe *et al.* Reactive oxygen species produced by the knockdown of manganese-superoxide dismutase up-regulate hypoxia-inducible factor-1 α expression in oral squamous cell carcinoma cells. *Free Radic Biol Med.* 48(10):1321-9 (2010).

Comment #2. In Figure 3, μ CT BV/TV data are shown twice (i.e., Figs. C & D). What are the differences? If there are differences, these need to be clarified better, please.

Response:

Figures 3c and 3d display the analysis of micro-CT and bone histomorphometry respectively. BV/TV in bone histomorphometry represents the proportion of bone volume within a two-dimensional (2D) sectioned slice of bone tissue, whereas in micro-CT, it provides a 3D analysis. Bone histomorphometry through H&E staining provides insights into cellular features such as the number of osteoblasts per bone perimeter (N.OB/B.Pm) and osteoblast surface per bone surface (Ob.S/BS). Conversely, micro-CT is more suitable for conducting non-destructive analysis on the 3D microarchitecture of bones, including bone mineral density (BMD), trabecular thickness (Tb.Th), trabecular separation (Tb.Sp), and trabecular number (Tb.N). In response to the reviewer's suggestion, we have included additional explanation to display BV/TV in both Figures 3c and 3d.

Results section (p. 7, Lines 151-152)

The **three-dimensional (3D)** microarchitecture of femoral trabecular bones in 4-month-old *Hif1a* conditional KO mice and their control *Hif1a^{fl/fl}* mice was analyzed using μ CT.

Results section (p. 7, Lines 157-160)

To complement the 3D- μ CT data, we conducted histomorphometric analysis of morphometric parameters of osteoblasts in the metaphyseal regions using H&E staining. It also revealed that the bone parameters, BV/TV, N.OB/B.Pm, and Ob.S/BS, showed higher values in *Hif1a^{fl/fl};Colla1-Cre* mice (Fig. 3d). Taken together, HIF-1 α is a negative regulator in osteoblast differentiation.

Comment #3. Figures 2 & 6: In the OVX HIF1 α knockdown model, what is the effect on HIF2 and Twist2 expression? What about RANKL expression and secretion?

Response:

In response to the reviewer's comments, we conducted additional experiments to investigate the effect on HIF-2 α and TWIST2 expression in the OVX *Hif1a* conditional knockout model. The representative IHC data have been displayed in Supplementary Figure 2 in the revised version of supplement data.

The results indicate that TWIST2 expression was reduced in OVX-*Hif1a^{fl/fl};Colla1-Cre* model compared to that of OVX-*Hif1a^{fl/fl}* control mice, while the expression of HIF-2 α and RANKL was just slightly decreased. Additionally, we confirmed expression of *Twist2* in siRNA-mediated silencing of *Hif1a* during primary osteoblast differentiation. This additional data is included in Figure 2i.

Results section (p. 7, Lines 141-142)

Moreover, we determined that the regulatory cis-elements of *Twist2* promoter contained putative HIF-1 α binding sequences, 5'-(A/G)CGTG-3', and ChIP results showed that *Twist2* is a direct target of HIF-1 α (Fig. 2h). **Additionally, siRNA-mediated silencing of *Hif1a* in primary cultured calvarial**

osteoblast confirmed that *Hif1a* regulates *Twist2* expression (Fig. 2i).

Results section (p. 9, Lines 206-209)

Bone histomorphometric analyses with H&E (Fig. 6b) and TRAP staining results (Fig. 6c) revealed that osteoblast-specific depletion of *Hif1a* prevented OVX-induced reduction of osteoblasts (Fig. 6b) but not OVX-induced bone resorption (Fig. 6c). In particular, osteoblast-specific depletion of *Hif1a* reduced TWIST2 expression in osteoblasts of the OVX model, while the effect of *Hif1a* deficiency on HIF-2 α and RANKL expression was not notably significant (Supplementary Fig. 2).

Comment #4. Figures 4 & 5: (1) what effect does HIF1 α over-expression have on osteoclast formation and resorption in vitro? (2) Could you provide the quantitative data alongside the qualitative Figure 5e, please?

Response:

(1) Regarding the comment about Fig 4

We appreciate the reviewer's constructive comments. In our previous report¹, HIF-2 α expression was increased during *in vitro* osteoclast differentiation of BMMs, whereas no significant differences in HIF-1 α expression were observed. We found that overexpression of HIF-2 α resulted in increased osteoclast formation and the number of multinucleated giant cells¹. However, our recent additional experiments determined that overexpression of HIF-1 α did not lead to any changes in osteoclast formation and resorption. The results are presented below but not included in the revised manuscript.

A. The mRNA levels of *Hif1a* and osteoclast-related genes (*Trap* and *Ctsk*) during RANKL-induced osteoclastogenesis of BMMs (n = 4). **B, C.** BMMs were infected with 400 MOI of Ad-C, Ad-*Hif1a*, or Ad-*Hif2a* on differentiation day 1 and then cultured with M-CSF and RANKL for 4 days. Trap staining (B) and quantification of multinucleated cells (C) were performed.

We have added the discussion in the context of the HIF-1 α and HIF-2 α on osteoclast formation and this could be the response to the reviewer's #8 comments, thus, we have stated this in the response #8.

(2) Regarding the comment about Fig 5

Following the reviewer's suggestion, we quantified the number of multinucleated giant cells and included it in Figure 5e. Additionally, we have added appropriate explanations in the 'Results' section.

Results section (p. 9, Lines 190-191)

Osteoclastogenesis of co-cultured BMMs with pre-osteoblasts transduced with Ad-*Hif2a* was more enhanced than co-culture with HIF-1 α overexpressing cells. This was further evidenced by an increase in the number of TRAP-positive multinucleated cells (Fig. 5e).

[Reference]

1. Lee *et al.* Controlling hypoxia-inducible factor-2 α is critical for maintaining bone homeostasis in mice. *Bone Res.* 7:14 (2019).

Comment #5. Figure 6a: When comparing Sham mice between fl/fl and *Coll1a1-Cre* mice (white vs. light grey dots), the increase in BMD, BV/TV, Tb.Th etc. between HIF1 α knockdown reported in Fig. 3a are no longer seen. I note that mice in Fig.3 are 4-months and mice used in Fig. 6 are 3 months old. Is this the main explanation for this discrepancy?

Response:

We appreciate the reviewer's critical comments. In our studies, we used male mice for all experiments except for the OVX models, where we utilized 4-month-old male mice for the experiments depicted in Fig. 3 and 3-month-old female mice for the OVX experiments depicted in Fig. 6. Although we did not explicitly describe the differences in bones between male and female mice in our original manuscript, several reports support the notion that male and female mice exhibit distinct bone strength and characteristics^{1,2,3}. Gender differences in bone have been associated with specific sex steroids and growth hormone levels in both animal and human studies^{1,2}. For example, Yao *et al.* demonstrated the gender-dependence of bone structure and properties³. The following figure is from the paper by Yao *et al.*, illustrating differences in certain bone parameters, such as trabecular bone BV/TV, trabecular bone number, trabecular bone separation, trabecular bone connectivity, and trabecular bone SMI, between wild-type female and male mice³. Furthermore, it shows different effects in female and male

osteogenesis imperfecta (OI) mouse models carrying a functional null mutation in the COL1A2 gene. According to the data presented in this paper, the effects of *Hif1a* deficiency on bone parameters were not observed in the sham groups of 3-month-old female mice in our study. We speculate that this discrepancy could be due to gender-dependent responses, but we acknowledge the need to consider other factors such as age differences (3-month-old female mice vs. 4-month-old male mice) and the inclusion of sham surgery in the OVX experimental condition. Further investigation is warranted, although we currently lack detailed insights. Therefore, we briefly discussed this in the 'Discussion' section of the revised manuscript. In addition, we added the gender information of the mice that we used in our study in the section of 'Materials and Methods' of the revised manuscript.

Discussion section (p. 14, Lines 317-323)

The results obtained in this study yielded somewhat challenging interpretations, among which is the difference in bone mass as depicted in Fig. 3 and Fig. 6. 4-month-old male *Hif1a^{fl/fl};Coll1a1-Cre* mice exhibited an increase in bone mass (Fig. 3), whereas 3-month-old female sham groups of *Hif1a^{fl/fl}* and *Hif1a^{fl/fl};Coll1a1-Cre* mice in OVX experiments showed no discrepancy in bone mass (Fig. 6a and b). One possible explanation for this inconsistency is gender-dependent differences in responses associated with specific sex steroids and growth hormone levels. However, further studies are required to clarify this aspect.

Materials and methods section (p. 17, 389-390)

Ethics statement and experimental mice

Male mice were used for all experiments except for the ovariectomized (OVX) osteoporosis models. For the OVX models, a 5-mm dorsal incision and sham operation (as a control) were performed using 8-week-old female mice.

[Reference]

1. Kim *et al.* The structural and hormonal basis of sex differences in peak appendicular bone strength in rats. *J Bone Miner. Res.* 18(1):150-155 (2003).
2. Martin, R. B. Size, structure and gender: lessons about fracture risk. *J Musculoskelet Neuronal Interact.* 2(3):209-211 (2002).
3. Yao *et al.* Gender-dependence of bone structure and properties in adult osteogenesis imperfecta murine model. *Ann Biomed Eng.* 41(6):1139-1149 (2013).

Comment #6. RNAseq methods & results section: please include the the day/time point that the sample was taken for RNAseq. Currently, this information is only provided in Fig.1 legend.

Response:

According to the reviewer's comment, we have included information regarding the preparation of RNA samples for RNAseq analysis in the 'Materials and Methods' section. Additionally, we have revised the legend for Fig. 1 to convey the information more clearly.

Materials and methods section (p. 19, Lines 433-434)

RNA sequencing (RNA-seq) analysis

For RNA-seq analysis, RNA samples were prepared from primary cultured calvarial pre-osteoblasts infected with control adenovirus (Ad-C), Ad-*Hif1a*, or Ad-*Hif2a*. Adenoviral infection was performed on day 3, followed by culturing the cells in osteogenic DM until day 6.

Comment #7. Methods line 360: Are the bone marrow macrophages freshly isolated and immediately cultured with primary osteoblasts?

Response:

In response to the reviewer's comment, we have added the method of isolation and primary culture of mouse bone marrow macrophages (BMM) in the 'Materials and Methods' section.

Materials and methods section (p. 19, Lines 418-425)

Co-culture of osteoblasts and bone marrow-derived macrophages (BMMs)

To perform the co-culture of osteoblasts and BMMs, primary calvarial pre-osteoblasts and BMMs were prepared as follows: Bone marrow was isolated from the long bones of 6- to 8-week-old mice and flushed with serum free α -MEM. Bone marrow cells were cultured in complete α -MEM for 24 h, after which non-adherent cells were collected and cultured in complete α -MEM in the presence of 30 ng/ml of M-CSF (Peprotech, Rocky Hill, NJ, USA) for 3 days. Adherent BMMs (2×10^4 cells per well in a 48-well plate) were then maintained in complete α -MEM containing 30 ng/ml of M-CSF for 24 h. Subsequently, they were co-cultured with primary calvarial pre-osteoblasts (4×10^3 cells/well) infected with Ad-C, Ad-*Hif1a*, or Ad-*Hif2a* in the presence of 100 ng/ml BMP-2, 50 μ g/ml L-AA, 5 mM β -Gp, and 10 nM 1,25-dihydroxyvitamin D₃ (VitD₃) for 5 days.

Comment #8. Discussion: I feel that there could be better contextualisation of the reported results to the wider literature. As one example, the relationship between the authors' previously published work on HIF2a could be included to link the current interplay between HIF1a and HIF2a, particularly in the context of osteoclast formation.

Response:

We appreciate the reviewer's constructive comments. This comment is essentially the similar to reviewer #2's comments. In response to the reviewers' feedback, we have included a discussion on the varied effects of HIFs on bone metabolism, along with a comparison of HIF-1 α and HIF-2 α regarding osteoclast formation, as mentioned in the reviewer's comment #4.

Discussion section (p. 13, Lines 279-287)

In our previous report, HIF-2 α expression was increased during in vitro osteoclast differentiation of BMMs, whereas no significant differences in HIF-1 α expression were observed. We found that overexpression of HIF-2 α resulted in increased osteoclast formation and the number of multinucleated giant cells. However, overexpression of HIF-1 α did not lead to any changes in osteoclast formation and resorption (data not shown). The effects of HIF-1 α on osteoclast formation and activation have been controversial. Some studies have observed an increase in osteoclast differentiation following HIF-1 α stimulation, while others have noted a decrease. Shirakura *et al.* and Wang *et al.* suggested the possibility that the increase in osteoclast differentiation under hypoxia is the result of crosstalk between osteoblasts and osteoclasts.

Comments from the reviewer #2:

This manuscript describes the effects of manipulation of HIF isoforms on osteoblastogenesis and osteoclastogenesis. The effects of Cre driven HIF deletion in osteoblasts on skeletal development is also assessed. Overall the manuscript describes a comprehensive set of experiments. However, especially in relation to in vivo experiments on transgenic mice, the effects of skeletal phenotype described are the opposite of what has been previously described in the literature. For example, the authors describe how deletion of HIF-1 α using a Col1a1 driver increases trabecular bone mass. Published studies using *Osx-cre* demonstrated reduced bone mass (PMCID: PMC4403258), *Bglap-cre* ; decreased bone volume (PMCID: PMC1878533 and PMID: 19899108) and *DMP-1 cre* no effect on bone volume (PMID: 30172741). Importantly, the Cre drivers used in these studies target deletion of HIF1 α in osteoblastic cell lineage cells both earlier and later than the Col1a1 driver.

The same is true regarding OVX studies in Figure 6. In this study deletion of HIF 1 α reduced the amount of bone loss induced by OVX. However, published literature shows that HIF 1 expression decreases with OVX (murine studies), and that *Vhl* deletion which upregulates HIF1 expression protects the skeleton from OVX-induced bone loss (PMID: 22193550).

Further – numerous studies using a variety of Cre-drivers demonstrate dramatically increased bone mass in mice with deletion of *Vhl* which upregulates expression/stabilization of both HIF isoforms. These published studies are not discussed and why the current study is in direct opposition is not addressed.

Response:

We appreciate the reviewer's constructive comment. Our results suggest that HIF-1 α is increased in mature osteoblasts during pathological conditions, resulting in decreased bone mass. Deletion of HIF-1 α in osteoblasts led to inhibition of bone loss caused by OVX and ageing. This finding is consistent with previous studies by Riddle and Frey, who suggested that mice lacking HIF-1 α in osteoblasts (using *Osteocalcin(Ocn)-Cre*) exhibit increased bone formation, and that HIF-1 α restricts the anabolic actions of PTH on bone remodeling, respectively¹.

It is well known that hypoxic-related conditions are associated with low bone mineral density, and hypoxic exposure increases the risk of bone fracture². Moreover, hypoxia has been noted to delay both osteoblast growth and differentiation, thereby limiting overall bone formation³. Because HIF- α family (HIF-1 α and HIF-2 α) is stabilized by hypoxic conditions, it is naturally expected that HIF- α is a factor that decreases bone density. This is consistent with our current findings suggesting a catabolic role of HIF-1 α in osteoblast function and bone homeostasis.

Hypoxic-induced inhibition of osteoblast function may also result of reduced PHD and lysyl oxidase enzyme activity^{3,4}. These oxygen-dependent enzymes are necessary for the posttranslational modification of collagen^{3,5}. Similarly, there is a report indicating that HIF-1 α reduces COL1A1 transcription through a distal promoter containing two GC boxes that bind Sp transcription factors⁶.

As mentioned by the reviewer, several researches reported that specific disruption of either HIF-1 α or HIF-2 α in osteoblasts disclosed similar roles in VEGF-mediated increase of skeletal vascularity whereas only HIF-1 α enhanced bone formation by regulating osteoblast differentiation and proliferation^{7,8,9}. In their experiments, osteoblast-specific *Hif1a* KO mice generated by crossing with

Ocn-Cre or *Osterix (Osx)-Cre* showed a decrease in bone mass, consistent with the study by Wang *et al.* suggesting that activation of the HIF-1 α pathway in osteoblasts of *Vhl* KO mice produces high levels of VEGF and leads to angiogenesis-dependent osteogenesis⁷. However, results from our experiments were distinct from those reported earlier. Osteoblast-specific *Hif2a* deletion in mice led to a significant increase in bone mass. This discrepancy may be explained in two ways.

First, we used *Cre*-transgenic mouse, *Type I collagen (Col1a1)-Cre*, to generate osteoblast-specific conditional KO mice whereas the other researchers used *Ocn-Cre*¹⁰, *Dmp-1-Cre*¹¹ or *Osx-Cre*¹² transgenic mice. *Col1a1* is expressed earlier than *Ocn* and *Dmp-1* during osteogenesis¹³. To further elucidate the expression pattern of osteogenic markers and *HIF-1a*, we examined *HIF1A*, *HIF2A*, *OCN* and *COL1A1* levels during osteogenesis in human mesenchymal stem cells. Consistent with results obtained using mouse pre-osteoblast cells (Supplementary Fig. 1a), mRNA expression patterns of *HIF1A* and *HIF2A* were similar to that of *COL1A1*, but *HIF1A* and *HIF2A* were gradually decreased at the *OCN*-expressing stage. This information supports the different phenotypes observed between our and earlier experiments. The results are presented below but not included in the revised manuscript.

In addition, conflicting results were reported by Wu *et al*¹⁴. They suggest that HIF-1 α is necessary within the context of combined PHD2 and PHD3 inactivation to inhibit bone resorption through an osteoblastic mechanism of OPG. Inconsistent results on the direct regulation of *Opg* by HIF-1 α was presented in our current study (Fig. 4a). In their report, *Osx-Cre* mice were used for generation of osteoblast specific conditional KO mice. The mouse *Osx-Cre* transgenic line expresses the *Cre* recombinase in committed osteoblast progenitors in both endochondral and membranous-derived bones¹⁵. Immature osteoblasts can be targeted by using the *Osx-Cre* transgenic mice, while mature osteoblasts can be targeted by using the *Col1a1-Cre* transgenic mice (specially the 2.3 kb *Col1a1-Cre* mice used in our experiment) based on their rich expression of type I collagen, the main constituent of bones. We used Type I collagen promoter (2.3 kb) *Col1a1-Cre* mice. The 3.6 kb *Col1a1-Cre* is expressed early during osteogenic differentiation, whereas the 2.3 kb *Col1a1-Cre* is activated later and its activity is restricted to maturing osteoblasts¹⁶. The 2.3 kb *Col1a1-Cre* is more suitable for our current study. These uncertain postulations should be further elucidated using various *Cre* mouse lines and multiple time-points (age). We have added a discussion about this issue in the discussion section.

Another possible explanation is the variable ages of mice used by the different groups. It has been suggested the HIF may exert differential effects depending on age and bone cycle (modeling vs. remodeling). Here, we used older mice (4 or 12 months old) relative to earlier groups (3, 6 or 12 weeks old)^{7,9} to evaluate the role of HIF-1 α in the bone remodeling process, and not bone ossification

(modeling). The angiogenesis-independent roles of HIF-1 α in bone remodeling remain unclear and should be further elucidated using various *Cre* mouse lines and multiple time-points (age).

Thus, this issue should be further investigated and is discussed in the revised manuscript to indicate this possibility.

Discussion section (p. 15, Lines 323-351)

Additionally, in our study, bone analyses were conducted using osteoblast-specific *Hif1 α* knockout mice, revealing different findings compared to those of previous studies. In our study, depletion of osteoblast-specific *Hif1 α* in mice resulted in a significant increase in bone mass through modulation of osteoblast functions. This inconsistency could be explained in two ways. Firstly, we utilized a different *Cre* transgenic model, *Coll1 α* -*Cre*, for generating osteoblast-specific conditional knockout mice, in contrast to the *Ocn*-*Cre* transgenic mice used by previous studies. *Coll1 α* is known to be expressed earlier than *Ocn* during osteogenesis. To further examine the expression patterns of osteogenic markers and HIF-1 α , we assessed *Hif1 α* , *Coll1 α* , and *Ocn* levels during osteogenesis in human mesenchymal stem cells (data not shown). Consistent with the results obtained using pre-osteoblast cells (Supplementary Fig. 1a), the mRNA expression pattern of *Hif1 α* closely resembled that of *Coll1 α* but decreased at the *Ocn*-expressing stage. In addition, conflicting results were reported by Wu *et al.* They suggest that HIF-1 α is necessary within the context of combined PHD2 and PHD3 inactivation to inhibit bone resorption through an osteoblastic mechanism of OPG. Inconsistent results on the direct regulation of *Opg* by HIF-1 α was presented in our current study (Fig. 4a). In their report, *Osterix (Osx)*-*Cre* mice were used for generation of osteoblast specific conditional KO mice. The mouse *Osx*-*Cre* transgenic line expresses the *Cre* recombinase in committed osteoblast progenitors in both endochondral and membranous-derived bones. Immature osteoblasts can be targeted by using the *Osx*-*Cre* transgenic mice, while mature osteoblasts can be targeted by using the *Coll1 α* -*Cre* transgenic mice (specially the 2.3 kb *Coll1 α* -*Cre* mice used in our experiment) based on their rich expression of type I collagen, the main constituent of bones. This finding supports the differing phenotypes observed between our study and earlier ones. Another possible explanation for the discrepant results could be the variations in the ages of mice used for experiments. It has been suggested the HIF may exert differential effects depending on age and bone cycle (modeling vs. remodeling) in our previous study. While previous studies analyzed juvenile or young adult mice, we utilized mature mice to assess the regulatory role of HIF-1 α in bone remodeling and osteoporotic bone loss. These hypotheses should be further investigated using various *Cre* mouse lines and multiple time-points (ages) in future studies.

[Reference]

1. Frey *et al.* Hypoxia-inducible factor-1 α restricts the anabolic actions of parathyroid hormone. *Bone Res.* 2:14005 (2014).
2. Wang *et al.* The hypobaric hypoxia environment impairs bone strength and quality in rats. *Int J Clin Exp Med.* . 10(6):9397-9406 (2017).
3. Utting *et al.* Hypoxia inhibits the growth, differentiation and bone-forming capacity of rat osteoblasts. *Exp Cell Res.* 312(10):1693-1702 (2006).
4. Arnett, T. R. Acidosis, hypoxia and bone. *Arch Biochem Biophys.* 503(1):103-109 (2010)
5. Myllyharju, J. Prolyl 4-hydroxylases, the key enzymes of collagen biosynthesis. *Matrix Biol.* 22(1):15-24 (2003).
6. Duval *et al.* Hypoxia inducible factor 1 alpha down-regulates type i collagen through Sp3 transcription factor in human chondrocytes. *IUBMB Life.* 68(9):756-763 (2016).
7. Wang *et al.* The hypoxia-inducible factor alpha pathway couples angiogenesis to osteogenesis during skeletal development. *J Clin Invest.* 117(6):1616-1626 (2007).
8. Wan *et al.* Activation of the hypoxia-inducible factor-1alpha pathway accelerates bone regeneration. *Proc Natl Acad Sci U S A.* 105(2):686-691 (2008).
9. Shomento *et al.* Hypoxia-inducible factors 1alpha and 2alpha exert both distinct and overlapping functions in long bone development. *J Cell Biochem.* 109(1):196-204 (2010).
10. Zhang *et al.* Osteoblast-specific knockout of the insulin-like growth factor (IGF) receptor gene reveals an essential role of IGF signaling in bone matrix mineralization. *J Biol Chem.* 277(46):44005-44012 (2002).
11. Lu *et al.* DMP1-targeted Cre expression in odontoblasts and osteocytes. *J Dent Res.* 86(4):320-325 (2007).
12. Rodda & McMahon. Distinct roles for Hedgehog and canonical Wnt signaling in specification, differentiation and maintenance of osteoblast progenitors. *Development.* 133(16):3231-3244 (2006).
13. Kim & Adachi. Cell Condensation Triggers the Differentiation of Osteoblast Precursor Cells to Osteocyte-Like Cells. *Front Bioeng Biotechnol.* 7:288 (2019).
14. Wu *et al.* Oxygen-sensing PHDs regulate bone homeostasis through the modulation of osteoprotegerin. *Genes Dev.* 29(8):817-831 (2015).
15. Nakashima *et al.* The novel zinc finger-containing transcription factor osterix is required for osteoblast differentiation and bone formation. *Cell.* 108(1):17-29 (2002).
16. Liu *et al.* Expression and activity of osteoblast-targeted Cre recombinase transgenes in murine skeletal tissues. *Int J Dev Biol.* 48(7):645-653 (2004).

Specific comments

Comment #1. Figure 1:

- (1) Data showing expression levels of HIFs at day 6, in figure 1a is not the same as that shown at the same timepoint in the supplementary figure 1. HIF1 and HIF2 are similar in Fig 1 but the supplementary figure suggests HIF2 expression exceeds HIF1.
- (2) Panel b shows immunostaining for HIF1 –similar data should be provided for HIF2.
- (3) Define normoxia in the context of your studies.

Response:

We appreciate the reviewer's meticulous comments.

(1) Regarding the comment about Fig. 1a and supplementary Fig. 1b

(mRNA expression) We have conducted additional analysis of expression of *Hif1a* and *Hif2a* in the early stage of osteoblast differentiation, and the results have been incorporated into Fig. 1a. It can be concluded that while there is some variation among experimental samples, the increased mRNA levels of *Hif1a* and *Hif2a* during osteoblast differentiation shows a similar level of enhancement.

(Protein expression) First, we need to correct the labeling mistake in supplementary Fig.1b. This figure illustrates the results of nuclear and cytosolic fractionation, but we inadvertently mislabeled the cytosolic fractions as whole lysates. Specifically, the data from the nuclear fraction indicated an increase in the translocation of both HIF-1 α and HIF-2 α into the nucleus by day 6, suggesting the possibility of their activation as transcription factors. Following the correction, we updated Supplementary Fig. 1b by rectifying the labeling of the cytosolic fraction and incorporating the data of HIF-1 α and HIF-2 α expression in whole lysates. Furthermore, it's important to note that the efficiency of the antibodies targeting HIF-1 α and HIF-2 α varied significantly. The antibody against HIF-2 α demonstrated excellent efficiency, whereas the one targeting HIF-1 α exhibited lesser efficacy. This discrepancy resulted in a higher density of bands for HIF-2 α compared to HIF-1 α in the Western blot results. The expression patterns of HIF-1 α and HIF-2 α at the protein level closely resembled those at the RNA level, as depicted in Supplementary Fig. 1a, and this alignment corroborated the data presented in Fig. 1a.

(2) Regarding the comment about panel b in Fig. 1

In response to the reviewer's comments, we have included the immunostaining data demonstrating the nuclear translocation of HIF-2 α in Figure 1b. The figure legend regarding the above results has also been revised.

Results section (p. 5, Lines 103-105)

Increased accumulation and nuclear localization of HIF-1 α and HIF-2 α were observed on day 6 with

the media containing differentiation-inducing agents, such as L-AA and β -Gp (Fig. 1b). Expression of HIF-1 α in nuclear persisted from day 6 to day 12 of differentiation, while HIF-2 α increased until day 15 (Supplementary Fig. 1b).

(3) Regarding the definition of normoxia in the context of our study

This comment is essentially the similar to Reviewer #1's comment 1. The term "Normoxia" refers to cells cultured under standard conditions of an ambient atmosphere supplemented with 5% CO₂. Typically, in cell culture models, this corresponds to an oxygen concentration of around 18%, given the controlled atmosphere of 95% air and 5% CO₂ in the incubator^{1,2}. This differs somewhat from the physiological oxygen environments where bone formation occurs, typically ranging from 12% to 5% oxygen concentration³. We have included an appropriate description of "Normoxia" in the 'Materials and Methods' section of the revised manuscript.

Materials & Methods (p. 18, Lines 397-400)

Primary culture of pre-osteoblasts and osteogenic differentiation

All experiments conducted in vitro culture in our study were under normoxic conditions. In general cell culture models, normoxic conditions are maintained in standard humidified cell culture incubators with 5% CO₂, where the oxygen concentration is approximately 18% O₂ (v/v), equivalent to an oxygen tension or partial pressure (pO₂) of 138 mmHg.

[Reference]

1. Wenger *et al.* Frequently asked questions in hypoxia research. *Hypoxia* (Auckl). 3:35-43 (2015).
2. Martinez *et al.* A Cell Culture Model that Mimics Physiological Tissue Oxygenation Using Oxygen-permeable Membranes. *Bio Protoc.* 9(18):e3371 (2019).
3. Utting *et al.* Hypoxia inhibits the growth, differentiation and bone-forming capacity of rat osteoblasts. *Exp Cell Res.* 312(10):1693-702 (2006).

Comment #2. Figure 2:

(1) Please clarify at what differentiation day target mRNAs were measured – text says cells were infected with adenovirus at day 3.

(2) The data is somewhat over interpreted. It is stated that Twist 2 knockdown restored the decrease of Ocn and Runx2 – this was a partial effect.

Response:

(1) Regarding the comment about differentiation day.

We apologize for the confusion caused by the lack of detailed explanation. This is now indicated and described in ‘Materials and Methods’ section.

Materials and Methods section (p. 18, Lines 412-416)

Adenoviral infection was performed on day 3 at the indicated MOI (Multiplicity of infection). For siRNA-mediated knockdown, primary cells were transfected with *Hif1a*-siRNA, *Hif2a*-siRNA, or *Twist2*-siRNA (Dharmacon, La Fayette, CO, USA) on differentiation day 3 using Lipofectamine RNAiMAX (Invitrogen, Carlsbad, CA, USA) following the manufacturer's recommendations. Non-targeting siRNA (scrambled; Bioneer, Daejeon, Korea) was used as a negative control. After gene overexpression and knockdown, cells were harvested on 6 days of osteoblast differentiation for further experiment.

(2) Regarding the comment about data overinterpretation

As pointed out by the reviewer, we agree that the data has been somewhat over interpreted. As suggested by the reviewer's opinion, considering that *Ocn* and *Runx2* are also regulated by other transcription factors, the partial restoration of the inhibition of these two genes by *Hif1a* overexpression through *Twist2* knockdown should be interpreted. Therefore we have revised this content in the ‘Results’ section.

Results section (p. 7, Lines 143-144)

Twist2 knockdown with specific siRNA partially restored the decrease of *Ocn* and *Runx2* by overexpression of *Hif1a* (Fig. 2j).

Comment #3. Figure 3. Please clarify where the bone images in panel a are taken from – provide a lower mag H&E image for localization.

Response:

In response to the reviewer's comments, we have included lower-magnification images for Fig. 3a. These images were stained with hematoxylin as a counterstain, and we believe they offer valuable localization information. We have annotated the positions of cells within the bone on the images using abbreviations to facilitate identification, and additional explanations have been included in the figure legend.

Comment #4. Figure 5. Text states that *Hif1a* expression is not affected by *Hif2a* overexpression – BUT panel d shows an upregulation of both *Hif1* and *Hif2* with Ad-*hif2a*. For panel e were osteoblasts transduced with Ad-*Hif2a* prior to co culture. The strong induction of osteoclastogenesis in 5d suggests HIF 2 is the driver.

Response:

We concur with the reviewer's comment. In the original Fig. 5d, it was difficult to compare the absolute increase in the expression of HIF-1 α and HIF-2 α . Therefore, to facilitate a direct comparison of the increases in HIF-1 α and HIF-2 α expression, the y-axis values of the graph were adjusted accordingly. In Fig. 5d, *Hif1a* exhibited a slight increase following the overexpression of *Hif2a*, averaging about 4-fold, which aligns with the RNA sequencing data demonstrating a 3.43-fold increase in Ad-*Hif2a*-mediated *Hif1a* expression (Fig. 1f). We inferred that the expression of *Hif1a* induced by *Hif2a* was relatively less than that of *Hif2a* induced by *Hif1a*, averaging about 14 times. However, upon considering the reviewer's point, it was deemed correct to analyze that the expression of *Hif1a* induced by *Hif2a* was also slightly increased, and this aspect has been corrected in the 'Results' section of the revised manuscript.

Results section (p. 8, Lines 176-177)

Consistent with the RNA-seq analysis, *Hif2a* expression was significantly increased by HIF-1 α overexpression, whereas *Hif1a* was slightly increased by HIF-2 α overexpression, while ectopic expression of each isoform of HIF- α showed an increase in *Rankl* expression (Fig. 5a).

General comments

Comment #5.

The size of the circles indicating individual data points are too large and obscure the bars and individual points themselves.

Response:

Following the reviewer's comment, we have improved all charts by adjusting the size of the circles representing individual data points to make them smaller.

Comment #6.

(1) There is a lack of detail in many of the figure legends. In many cases it is unclear how long cells were cultured in osteogenic conditions.

(2) Further, the authors should provide data to show how quickly Ad-HIF transfection increases HIF1 or HIF2 and how long the effects are sustained. The same criticism can be leveled at siRNA approaches. In many experiments, cells are cultured for extended periods of time and it is unknown whether effects on HIFs are sustained.

Response:

(1) As requested by the reviewer, detailed information has been added to the figure legends for Fig. 1, Fig. 2, Fig. 4, and Fig 5 in the revised manuscript.

(2) In our study, *Hif1a* adenovirus (Ad-*Hif1a*) and *Hif1a* siRNA were infected or transfected on day 3 and harvested on day 6 during osteoblast differentiation. In these experiments, sufficient overexpression or knockdown of *Hif2a* by Ad-*Hif1a* or *Hif1a* siRNA, respectively, was examined by qPCR and western blotting. As an additional experiment aligned with the objective of our study (in response to the reviewer's question), we conducted experiments where infection or transfection was carried out for only 2 days. The results revealed that the overexpression and knockdown of *Hif1a* were more pronounced in samples harvested 2 days after infection (transfection) compared to those harvested 3 days after infection (transfection), as used in our main experiments. However, considering the main goal of our study, which is to assess osteoblast differentiation and the expression of osteoblast markers, clearer results were obtained from samples harvested 3 days after treatment. Therefore, we consider the process of our experiment to be appropriate.

Comment #7.

Please check all references for applicability and that they have been cited correctly. For example, there is no mention of hypoxia or HIFs in some of these references “Remarkably, HIF- α can be stabilized even in normoxia by inflammatory factors, such as interleukin (IL)-70 1 β , IL-6, IL-17, and tumor necrosis factor (TNF)- α , in a pathophysiologic microenvironment, and normoxic stabilization of HIF-1 α induces osteoclastogenesis and pathological bone resorption^{11–16}.”

Further this interpretation of this reference is incorrect "Loss of HIF-1a is responsible for the increased accumulation of bone matrix" line 81.

Response:

We apologize for the confusion caused by the mistake. We have verified that all references have been correctly cited again. And, we have modified that sentence.

Introduction section (p. 4, Lines 78-79)

On the contrary, other studies support the catabolic effects of HIF-1 α . **Deletion of HIF-1 α in osteoblasts of mature bone resulted in the increased accumulation of bone.** Overexpression of ~

Comment #8.

This manuscript should be thoroughly checked for grammar. In many cases grammatical errors impact the clarity of the message. The following sentence in the abstract is one such example “Our findings conclude that HIF-1 α plays a crucial role in regulating bone homeostasis, such as osteoblast differentiation independently on HIF-2 α and osteoclast activation via crosstalk with osteoblasts depending on HIF-2 α .”.

Response:

We apologize for any confusion caused by grammatical errors. We have thoroughly reviewed the manuscript for such errors in accordance with the reviewer's feedback, including the abstract.

Abstract section (p. 2, Lines 40-43)

Our findings conclude that HIF-1 α plays an important role in regulating bone homeostasis by controlling osteoblast differentiation, and in influencing osteoclast formation through the regulation of RANKL secretion via HIF-2 α modulation.

Reviewers' comments:

Reviewer #1 (Remarks to the Author):

Thank you, once again, for the opportunity to review this manuscript. The authors have sufficiently addressed my previous comments.

My outstanding question, upon re-reviewing the manuscript, is why is over-expression of HIF-1a and -2a (i.e., Ad-HIF1a and Ad-HIF2a) used for the RNAseq if the authors are 'modelling' normoxic stabilisation? The authors indicate that these HIFs are stabilised under normoxic conditions; therefore, why not study the differential expression under standard conditions (i.e., no induction of over-expression)?

Reviewer #2 (Remarks to the Author):

The authors have adequately addressed all comments.

Point-by-point responses to reviewers' comments

Comments from the reviewer #1:

Thank you, once again, for the opportunity to review this manuscript. The authors have sufficiently addressed my previous comments.

My outstanding question, upon re-reviewing the manuscript, is why is over-expression of HIF-1 α and -2 α (i.e., Ad-HIF1 α and Ad-HIF2 α) used for the RNAseq if the authors are 'modelling' normoxic stabilisation? The authors indicate that these HIFs are stabilised under normoxic conditions; therefore, why not study the differential expression under standard conditions (i.e., no induction of over-expression)?

Response:

We appreciate the reviewer's comments. Previous work, investigating the function of HIF-2 α ¹, and the current study on HIF-1 α , suggest that the increase in both HIF-1 α and HIF-2 α during osteoblast differentiation allows for the regulation of their target gene expression, thereby negatively regulating osteoblast differentiation. The primary purpose of RNAseq in this study was to screen and profile target-candidate genes of each HIF-1 α and HIF-2 α as transcription factors, with the secondary purpose being to compare the target gene profiles.

In our study, we observed the accumulation of HIF-1 α and HIF-2 α during osteoblast differentiation under normoxia. This increase in HIF-1 α and HIF-2 α was attributed to the upregulation of their gene expression, as determined by RT-PCR and qRT-PCR. Additionally, it could be the result of the stabilization of their proteins by escaping proteasomal degradation.

From the reviewer's perspective, investigating the target gene profile through HIF stabilization under normoxia could be a meaningful approach. For instance, gene expression profiling by HIF stabilization could be achieved through knock-down of PHD (HIF-prolyl hydroxylase domain protein), which hydroxylates HIFs, or knock-down of VHL (von-Hippel Lindau tumor suppressor), a component of an E3 ubiquitin ligase complex. However, this approach may not be suitable for profiling and comparing target genes of each isoform of HIFs since it stabilizes both HIF-1 α and HIF-2 α . Additionally, it may lead to the stabilization of other proteins besides HIFs, making it difficult to confirm specific effects solely attributable to HIFs². To assess the differential effects of stabilization of each isoform of HIF- α , knock-down of HIF-1 α and HIF-2 α using differentiated osteoblasts could be considered for RNA-seq analysis. However, this knock-down method might be relatively indirect compared to the overexpression method when considering the profiling and comparison of target genes of HIF-1 α and HIF-2 α as transcription factors. Consequently, we chose the overexpression method for RNA-seq analysis.

Incorporating the feedback from the reviewer, we have added this aspect in the ‘Discussion’ part in the revised manuscript.

Discussion section (p. 12, Lines 267-269)

The RNA-seq analysis to determine gene expression profile by HIF-1 α and HIF-2 α overexpression during osteoblast differentiation (Fig. 1) showed that about 73.3% DEGs of a total of 23,997 genes were similarly regulated, and about 6% and 20.7% DEGs were differentially regulated by HIF-1 α and HIF-2 α , respectively (Fig. 1d). **Performing additional experiments comparing the RNA-seq results from osteoblasts with knockdown of each HIF isoform would provide more robust evidence for conclusive insights.**

[Reference]

1. Lee *et al.* Controlling hypoxia-inducible factor-2 α is critical for maintaining bone homeostasis in mice. *Bone Res.* 7:14 (2019).
2. Jung *et al.* Estrogen receptor α is a novel target of the Von Hippel-Lindau protein and is responsible for the proliferation of VHL-deficient cells under hypoxic conditions. *Cell Cycle.* 11(23):4462-4473 (2012)

REVIEWERS' COMMENTS:

Reviewer #1 (Remarks to the Author):

Many thanks for addressing my question. Happy for this manuscript to be published - congratulations.